# JointDiff: Bridging Continuous and Discrete in Multi-Agent Trajectory Generation

**Guillem Capellera**[1,2,3]    **Luis Ferraz**[1]    **Antonio Rubio**[1]    **Alexandre Alahi**[2]    **Antonio Agudo**[3]

[1] Kognia Sports Intelligence    [2] Visual Intelligence for Transportation, EPFL[*]
[3] Institut de Robòtica i Informàtica Industrial, CSIC-UPC

## Abstract

Generative models often treat continuous data and discrete events as separate processes, creating a gap in modeling complex systems where they interact synchronously. To bridge this gap, we introduce **JointDiff**, a novel diffusion framework designed to unify these two processes by simultaneously generating continuous spatio-temporal data and synchronous discrete events. We demonstrate its efficacy in the sports domain by simultaneously modeling multi-agent trajectories and key possession events. This joint modeling is validated with non-controllable generation and two novel controllable generation scenarios: *weak-possessor-guidance*, which offers flexible semantic control over game dynamics through a simple list of intended ball possessors, and *text-guidance*, which enables fine-grained, language-driven generation. To enable the conditioning with these guidance signals, we introduce **CrossGuid**, an effective conditioning operation for multi-agent domains. We also share a new unified sports benchmark enhanced with textual descriptions for soccer and football datasets. JointDiff achieves state-of-the-art performance, demonstrating that joint modeling is crucial for building realistic and controllable generative models for interactive systems. Project

## 1 Introduction

Modeling the dynamics of multi-agent systems is fundamentally challenging when continuous motion is tightly coupled with discrete, state-altering events. This interplay is critical in domains like autonomous driving and robotics, but finds a particularly rich and demanding testbed in team sports. Here, the continuous trajectories of players are synchronously intertwined with discrete events like passes and possessions. Generating realistic sports gameplay therefore requires a model that can jointly represent these two modalities. However, existing generative models often fall short by treating these components in isolation. This can lead to physically implausible generations, such as unrealistic passes or flawed ball-possessor interactions (Lee et al., 2024; Capellera et al., 2025). While deterministic models have started to incorporate events (Kim et al., 2023; Capellera et al., 2024), a comprehensive generative framework is missing. This deficiency is compounded by evaluation protocols that rely on individual-level metrics like minimum ADE/FDE (Alahi et al., 2016), which were inherited from pedestrian forecasting and fail to capture scene-level coherence (Casas et al., 2020; Girgis et al., 2021; Weng et al., 2023; Capellera et al., 2025), crucial to team sports.

To address this gap, we turn to the expressive power of diffusion models. While continuous diffusion (Ho et al., 2020) has excelled at generating high-fidelity data like trajectories (Mao et al., 2023; Jiang et al., 2023; Gu et al., 2022; Rempe et al., 2023; Bae et al., 2024; Li et al., 2023; Yang et al., 2024; Capellera et al., 2025), discrete diffusion (Hoogeboom et al., 2021; Austin et al., 2021) has concurrently emerged as a potent, non-autoregressive alternative to large language models (LLMs) for structured sequence generation (Lou et al., 2023). Nascent work has begun to unify these modalities for static tasks such as layout design (Levi et al., 2023) and visual-language modeling (Li et al., 2025). Our key insight is to unify these two paradigms for the temporally evolving complex systems. We introduce JointDiff, a novel framework that, to the best of our knowledge, is the first to apply joint continuous-discrete diffusion to simultaneously generate spatio-temporal continuous data (trajectories) alongside its corresponding synchronous temporal discrete events (possession events).

---

[*]This work was performed during a research stay at EPFL, Laussanne (Switzerland).
Contact:{firstname.lastname}@{kogniasports.com,epfl.ch,upc.edu}

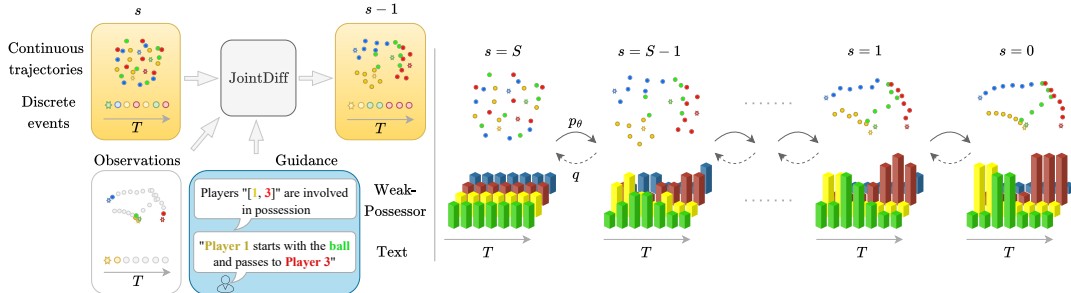

Figure 1: **JointDiff**. Our model jointly generates continuous trajectories and discrete events, with guidance provided through either weak-possessor information or natural language text. Stars (★) refer to the initial timestep.

Beyond realism, a truly useful generative model also needs to be controllable. While diffusion-guidance methods (Dhariwal & Nichol, 2021; Ho & Salimans, 2022) have been used in motion synthesis to satisfy pedestrian goals or constraints (Jiang et al., 2023; Rempe et al., 2023), semantic control through discrete events in multi-agent systems remains unexplored. Our joint framework directly enables such control using the classifier-free guidance (CFG) (Ho & Salimans, 2022). We introduce weak-possessor-guidance (WPG), a novel conditioning method that allows users to steer gameplay by simply providing an ordered list of intended ball possessors, without rigid timing constraints. We further extend controllability to natural language via text-guided generation, facilitated by a new, curated benchmark of text descriptions for soccer and football datasets. Our approach is illustrated in Fig. 1. In summary, our principal contributions are: **1)** A novel joint continuous-discrete diffusion framework that simultaneously generates multi-agent trajectories and synchronous discrete events, leading to more realistic and coherent scenes; **2)** Enabling high-level semantic controllability in dynamic domains. We introduce two novel controllable tasks (weak-possessor-guidance and text-guidance) and a dedicated CrossGuid module that effectively injects conditioning signals into the structured multi-agent embedding; **3)** A unified benchmark for multi-agent modeling in sports, enhanced with new text descriptions for soccer and football datasets. Our method achieves state-of-the-art results on scene-level metrics.

## 2 RELATED WORK

**Trajectory Modeling.** The evolution of multi-agent trajectory modeling has progressed from Recurrent Neural Networks (RNNs) and Variational RNNs (VRNNs) (Zheng et al., 2016; Felsen et al., 2018; Zhan et al., 2019; Yeh et al., 2019; Li et al., 2021), to generative models like Generative Adversarial Networks (GANs) (Gupta et al., 2018; Sadeghian et al., 2019; Fang et al., 2020) and Conditional Variational Autoencoders (CVAEs) (Salzmann et al., 2020; Graber & Schwing, 2020; Yuan et al., 2021; Lee et al., 2022; Xu et al., 2022; Zheng et al., 2024). In recent years, non-sampling approaches built on transformers (Vaswani et al., 2017), and in some cases enriched with visual data (Saadatnejad et al., 2023; Gao et al., 2024; Wu et al., 2024), have achieved notable progress in multimodal future prediction by effectively modeling long-range spatio-temporal dependencies (Alcorn & Nguyen, 2021; Girgis et al., 2021; Ngiam et al., 2021; Lee et al., 2024). Building on this progress, Denoising Diffusion Probabilistic Models (DDPMs) (Ho et al., 2020) have emerged as the state-of-the-art for generating high-fidelity and diverse trajectories (Mao et al., 2023; Jiang et al., 2023; Gu et al., 2022; Rempe et al., 2023; Bae et al., 2024; Li et al., 2023; Fu et al., 2025). This generative power also extends to tasks like trajectory completion, where the recent diffusion model U2Diff (Capellera et al., 2025) has surpassed prior methods based on Graph VRNNs (GVRNNs) (Omidshafiei et al., 2022; Xu et al., 2023), GANs (Liu et al., 2019), and CVAEs (Xu & Fu, 2025). Notably, U2Diff also competes against forecasting-specific architectures despite using an Independent and Identically Distributed (IID) sampling method and without using time window constraints.

**Multi-agent Controllability.** Diffusion models have recently been augmented with guided sampling to satisfy user-specified constraints or objectives. Existing pedestrian and autonomous driving methods (Rempe et al., 2023; Jiang et al., 2023; Yang et al., 2024), typically focus on controlling individual-level attributes like waypoints, speeds, or physics constraints. Similarly, human motion generation approaches (Karunratanakul et al., 2023) and robotics planning methods (Mishra et al.,

2023; Fang et al., 2024) often guide a single agent. In contrast, our work focuses on controlling a broader multi-agent system through high-level semantic directives. We adopt the CFG paradigm (Ho & Salimans, 2022), widely used in image and video (Rombach et al., 2022; Ho et al., 2022), to bias generation toward a user-specified sequence of possessors or a natural language description. This allows for a comprehensive control of the entire scene rather than individual agent behavior.

**Joint Continuous-Discrete Diffusion**. Joint diffusion models for mixed continuous–discrete data are an emerging research direction, with applications in static domains such as layout design (Levi et al., 2023), CAD sketches (Chereddy & Femiani, 2025), and vision–language modeling (Li et al., 2025), where absorbing state diffusion (Austin et al., 2021) is commonly used for discrete variables. In contrast, dynamic domains have been underexplored. Prior work (Zeng et al., 2024) applies the multinomial formulation (Hoogeboom et al., 2021) to temporal point processes, but it's restricted to single-instance future prediction and relies on sequential, non-simultaneous generation. We extend the multinomial formulation to general controllable dynamic domains, exemplified by the multi-agent completion task, and introduce a unified diffusion framework that simultaneously models continuous trajectories and discrete events. This formulation proves more consistent than absorbing state diffusion in our temporally evolving domain, as it enables continuous refinement of discrete variables throughout the denoising process. Furthermore, we are first to incorporate high-level semantic controllability, such as WPG and text-guidance, for joint continuous–discrete generation in dynamic domains, consistently outperforming the non-joint baseline.

## 3 DIFFUSION BACKGROUND

Diffusion models are a class of generative models that learn to reverse a progressive noising process, operating in two stages: a *forward diffusion* process and a learnable *reverse denoising* process. The *forward process* is a fixed Markov chain that gradually adds noise to a data sample $\mathbf{X}_0 \sim q(\mathbf{X}_0)$. Over $S$ steps, the data is corrupted following a variance schedule, $\{\beta_s \in (0, 1)\}_{s=1}^{S}$, until $p(\mathbf{X}_S)$ resembles a simple and known noise distribution. This process is defined as: $q(\mathbf{X}_{1:S} \mid \mathbf{X}_0) = \prod_{s=1}^{S} q(\mathbf{X}_s \mid \mathbf{X}_{s-1})$. A key property is that we can sample $\mathbf{X}_s$ at any arbitrary timestep conditioned on the initial data $\mathbf{X}_0$ in a closed form $q(\mathbf{X}_s \mid \mathbf{X}_0)$. The *reverse process* is a generative model that learns to denoise the data by iteratively reversing the forward steps. Starting with a sample from the known noise distribution, $\mathbf{X}_S$, a neural network, $p_\theta$, learns to approximate the reverse transitions: $p_\theta(\mathbf{X}_{0:S}) = p(\mathbf{X}_S) \prod_{s=1}^{S} p_\theta(\mathbf{X}_{s-1} \mid \mathbf{X}_s)$.

The objective is to train the model to generate new samples that match the original data distribution $q(\mathbf{X}_0)$. This is achieved by minimizing a variational upper bound on the negative log-likelihood:

$$\mathcal{L}_{\text{vb}} = \mathbb{E}_q[-\log p_\theta(\mathbf{X}_0 \mid \mathbf{X}_1)] + \sum_{s=2}^{S} \mathbb{E}_q[D_{\text{KL}}(q(\mathbf{X}_{s-1} \mid \mathbf{X}_s, \mathbf{X}_0) \,\|\, p_\theta(\mathbf{X}_{s-1} \mid \mathbf{X}_s))] + \text{C}, \quad (1)$$

where C is a constant term defined by $D_{\text{KL}}(q(\mathbf{X}_S \mid \mathbf{X}_0) \,\|\, p(\mathbf{X}_S))$. The true posterior $q(\mathbf{X}_{s-1} \mid \mathbf{X}_s, \mathbf{X}_0)$ is tractable, which allows for a direct optimization approach of the neural network approximating the reverse transition $p_\theta(\mathbf{X}_{s-1} \mid \mathbf{X}_s)$.

While the original DDPM framework (Ho et al., 2020) handled continuous data with Gaussian noise, subsequent work adapted it for discrete, categorical data (Hoogeboom et al., 2021; Austin et al., 2021). Our work builds on multinomial diffusion, which corrupts discrete events toward a uniform distribution (see Appendix A for details on these frameworks and their loss terms).

## 4 METHOD

### 4.1 PROBLEM STATEMENT

We model a dynamic scene composed of $N$ agents (e.g., ball and players) over a time horizon of $T$ timesteps. The state of the scene at any time $t$ is described by a combination of continuous and discrete variables as:

- Continuous state: The agent's 2D spatial coordinates are represented by a tensor $\mathbf{Y} \in \mathbb{R}^{T \times N \times 2}$, where $y_{t,n} \in \mathbb{R}^2$ is the position of agent $n$ at time $t$. The agents are indexed such that $n = 0$ refers to the ball, and $n = \{1, \ldots, N-1\}$ refers to the players.

- Discrete state: A categorical event at each timestep, such as ball possession, is represented by a one-hot matrix $\mathbf{E} \in \{0, 1\}^{T \times N}$. Each row $\mathbf{e}_t$ is a one-hot vector where a value of 1 at index $n$ indicates that agent $n$ is in possession of the ball. States $\mathbf{e}_t$ where the ball is not possessed (e.g., during a pass or shot) are assigned to the ball's own category, index $n = 0$.

The complete scene is described by a tuple $\mathbf{X} = (\mathbf{Y}, \mathbf{E})$, which jointly represents the spatio-temporal trajectories and discrete events. We define two generative objectives for our model: **Completion and Controllable Generation.** *The first objective* is to generate plausible and coherent completions of a dynamic scene. Given a set of partial observations $\mathbf{X}^{\mathrm{co}} = (\mathbf{Y}^{\mathrm{co}}, \mathbf{E}^{\mathrm{co}})$ defined by a binary mask $\mathbf{M}$ that specifies which time steps and agents are observed, the goal is to learn a model capable of sampling from the conditional distribution $p(\mathbf{X} \mid \mathbf{X}^{\mathrm{co}})$. *The second objective* extends this to controllable generation by introducing an external conditioning variable $\mathcal{G}$ (e.g., natural language text) to guide the generation process. The model must learn to sample from the augmented conditional distribution $p(\mathbf{X} \mid \mathbf{X}^{\mathrm{co}}, \mathcal{G})$. This framework enables scene generation that is influenced not only by partial observations but also by additional data from different domains.

## 4.2 Joint Continuous-Discrete Diffusion

To model the joint data distribution $q(\mathbf{X}_0) = q(\mathbf{Y}_0, \mathbf{E}_0)$, we design a diffusion model that simultaneously handles both continuous trajectories and discrete events.

The forward process corrupts the initial data $\mathbf{X}_0 = (\mathbf{Y}_0, \mathbf{E}_0)$ over $S$ timesteps. We assume the noising processes for the two modalities are independent, which allows us to factorize the joint forward transition as:

$$q(\mathbf{Y}_s, \mathbf{E}_s \mid \mathbf{Y}_0, \mathbf{E}_0) = q(\mathbf{Y}_s \mid \mathbf{Y}_0) \, q(\mathbf{E}_s \mid \mathbf{E}_0). \tag{2}$$

This factorization enables the application of a continuous diffusion process to the trajectories $\mathbf{Y}_0$ and a discrete diffusion process to the events $\mathbf{E}_0$. For simplicity, we assume both processes are governed by a shared variance schedule $\{\beta_s\}_{s=1}^S$. The individual closed-form transitions are:

$$q(\mathbf{Y}_s \mid \mathbf{Y}_0) = \mathcal{N}(\mathbf{Y}_s; \sqrt{\bar{\alpha}_s}\mathbf{Y}_0, (1 - \bar{\alpha}_s)\mathbf{I}), \tag{3}$$

$$q(\mathbf{E}_s \mid \mathbf{E}_0) = \mathrm{Cat}(\mathbf{E}_s; \bar{\alpha}_s \mathbf{E}_0 + (1 - \bar{\alpha}_s)/N). \tag{4}$$

where $\alpha_s = 1 - \beta_s$ and $\bar{\alpha}_s = \prod_{i=1}^s \alpha_i$, and $\mathrm{Cat}(; p)$ denotes a categorical distribution with probabilities $p$. Equation 3 describes a standard Gaussian diffusion process from DDPM (Ho et al., 2020), where the initial state $\mathbf{Y}_0$ is progressively corrupted with Gaussian noise. Following Hoogeboom et al. (2021), Eq. 4 defines a multinomial diffusion process, where the one-hot matrices $\mathbf{E}_0$ are gradually mixed with a uniform distribution over $N$ categories. As $s \to S$, $\mathbf{Y}_s$ converges to a sample from an isotropic Gaussian, and $\mathbf{E}_s$ to a sample from a uniform categorical distribution.

For the reverse process, we make the conditional independence assumption at $s - 1$, allowing the joint posterior to be factorized as:

$$p_\theta(\mathbf{Y}_{s-1}, \mathbf{E}_{s-1} \mid \mathbf{Y}_s, \mathbf{E}_s, \mathbf{X}^{\mathrm{co}}, \mathcal{G}) = p_\theta(\mathbf{Y}_{s-1} \mid \mathbf{Y}_s, \mathbf{E}_s, \mathbf{X}^{\mathrm{co}}, \mathcal{G}) \, p_\theta(\mathbf{E}_{s-1} \mid \mathbf{Y}_s, \mathbf{E}_s, \mathbf{X}^{\mathrm{co}}, \mathcal{G}). \tag{5}$$

Note that the model learns the dependencies between the continuous and the discrete modalities because the reverse network $p_\theta$ is conditioned on the full state $\mathbf{X}_s = (\mathbf{Y}_s, \mathbf{E}_s)$. The reverse process is parametrized with a single neural network with two prediction heads. The network takes the noisy state $(\mathbf{Y}_s, \mathbf{E}_s)$, the denoising step $s$, the partial observations $\mathbf{X}^{\mathrm{co}}$, and optional guidance $\mathcal{G}$ as input.

- A regression head predicts the noise added to the trajectories, denoted as $\epsilon_\theta(\mathbf{Y}_s, \mathbf{E}_s, s, \mathbf{X}^{\mathrm{co}}, \mathcal{G})$.

- A classification head predicts the original event probabilities, $\hat{\mathbf{E}}_0 = \pi_\theta(\mathbf{Y}_s, \mathbf{E}_s, s, \mathbf{X}^{\mathrm{co}}, \mathcal{G})$.

The **continuous** reverse transition is defined as a Gaussian distribution:

$$p_\theta(\mathbf{Y}_{s-1} \mid \mathbf{Y}_s, \mathbf{E}_s, \mathbf{X}^{\mathrm{co}}, \mathcal{G}) = \mathcal{N}(\mathbf{Y}_{s-1}; \boldsymbol{\mu}_\theta(\mathbf{Y}_s, \mathbf{E}_s, s, \mathbf{X}^{\mathrm{co}}, \mathcal{G}), \sigma_s^2 \mathbf{I}), \tag{6}$$

where the variance $\sigma_s^2$ is a non learnable hyperparameter, typically set to $\frac{1 - \bar{\alpha}_{s-1}}{1 - \bar{\alpha}_s}\beta_s$, and the mean $\boldsymbol{\mu}_\theta$ is computed from the predicted noise $\epsilon_\theta$ using the standard DDPM parametrization as $\boldsymbol{\mu}_\theta(\mathbf{Y}_s, s) = \frac{1}{\sqrt{\alpha_s}}\left(\mathbf{Y}_s - \frac{\beta_s}{\sqrt{1 - \bar{\alpha}_s}}\epsilon_\theta(\mathbf{Y}_s, s)\right)$.

The **discrete** reverse transition is derived by plugging the network's prediction $\hat{\mathbf{E}}_0$ into the true posterior $q(\mathbf{E}_{s-1} \mid \mathbf{E}_s, \mathbf{E}_0)$ for steps $s \geq 2$, while for the final step $(s = 1)$ we directly use the categorical distribution with parameter $\hat{\mathbf{E}}_0$. Specifically, we have for $s = 1$ and $s \geq 2$, respectively:

$$p_\theta(\mathbf{E}_0 \mid \mathbf{Y}_1, \mathbf{E}_1, \mathbf{X}^{\text{co}}, \mathcal{G}) = \text{Cat}(\mathbf{E}_0; \hat{\mathbf{E}}_0) \quad \text{and} \quad p_\theta(\mathbf{E}_{s-1} \mid \mathbf{Y}_s, \mathbf{E}_s, \mathbf{X}^{\text{co}}, \mathcal{G}) = q(\mathbf{E}_{s-1} \mid \mathbf{E}_s, \hat{\mathbf{E}}_0).$$

The posterior is a categorical distribution $q(\mathbf{E}_{s-1} \mid \mathbf{E}_s, \mathbf{E}_0) = \text{Cat}(\mathbf{E}_{s-1}; , \boldsymbol{\theta}_{\text{post}}(\mathbf{E}_s, \mathbf{E}_0))$ whose probabilities are defined as:

$$\boldsymbol{\theta}_{\text{post}}(\mathbf{E}_s, \mathbf{E}_0) = \tilde{\boldsymbol{\theta}} / \sum_{n=0}^{N-1} \tilde{\boldsymbol{\theta}}_n \quad \text{and} \quad \tilde{\boldsymbol{\theta}} = [\alpha_s \mathbf{E}_s + (1 - \alpha_s)/N] \odot [\bar{\alpha}_{s-1} \mathbf{E}_0 + (1 - \bar{\alpha}_{s-1})/N].$$

**Training Objective.** Our model is trained end-to-end by minimizing a joint objective derived from Eq. 1. Since the forward process $q$ acts independently on each modality, the true posterior also factorizes:

$$q(\mathbf{X}_{s-1} \mid \mathbf{X}_s, \mathbf{X}_0) = q(\mathbf{Y}_{s-1} \mid \mathbf{Y}_s, \mathbf{Y}_0) \, q(\mathbf{E}_{s-1} \mid \mathbf{E}_s, \mathbf{E}_0). \tag{7}$$

The key property of the KL divergence is that it decomposes over factorized distributions. This allows the variational bound from Eq. 1 to be separated into continuous and discrete terms:

$$\mathcal{L}_{\text{vb}} = \mathbb{E}_q \Big[ - \log p_\theta(\mathbf{Y}_0 \mid \mathbf{Y}_1, \mathbf{E}_1, \mathbf{X}^{\text{co}}, \mathcal{G}) + \sum_{s=2}^{S} D_{\text{KL}}\big(q(\mathbf{Y}_{s-1} \mid \mathbf{Y}_s, \mathbf{Y}_0) \,\|\, p_\theta(\mathbf{Y}_{s-1} \mid \mathbf{Y}_s, \mathbf{E}_s, \mathbf{X}^{\text{co}}, \mathcal{G})\big) \Big]$$

$$+ \mathbb{E}_q \Big[ - \log p_\theta(\mathbf{E}_0 \mid \mathbf{Y}_1, \mathbf{E}_1, \mathbf{X}^{\text{co}}, \mathcal{G}) + \sum_{s=2}^{S} D_{\text{KL}}\big(q(\mathbf{E}_{s-1} \mid \mathbf{E}_s, \mathbf{E}_0) \,\|\, p_\theta(\mathbf{E}_{s-1} \mid \mathbf{Y}_s, \mathbf{E}_s, \mathbf{X}^{\text{co}}, \mathcal{G})\big) \Big] = \mathcal{L}_{\text{vb}}^{\mathbf{Y}} + \mathcal{L}_{\text{vb}}^{\mathbf{E}}$$

For the continuous part, we use the simplified objective common in DDPMs, reducing the objective $\mathcal{L}_{\text{vb}}^{\mathbf{Y}}$ to $\mathcal{L}_{\text{simple}}^{\mathbf{Y}} = \mathbb{E}_{s, \mathbf{X}_0, \epsilon} \big[ \|\epsilon - \epsilon_\theta(\mathbf{Y}_s, \mathbf{E}_s, s, \mathbf{X}^{\text{co}}, \mathcal{G})\|_2^2 \big]$, where $\epsilon$ is the Gaussian noise injected at step $s$. For the discrete modality, we retain the exact variational form $\mathcal{L}_{\text{vb}}^{\mathbf{E}}$ (see Appendix A for more details on how to compute each loss term). Our proposed resulting training objective is the weighted combination:

$$\mathcal{L}_{\text{joint}} = \mathcal{L}_{\text{simple}}^{\mathbf{Y}} + \lambda \, \mathcal{L}_{\text{vb}}^{\mathbf{E}}, \tag{8}$$

where $\lambda$ is a balancing hyperparameter chosen so that both modalities contribute comparably during optimization. Instead of uniform sampling, we use the importance sampling method proposed by Nichol & Dhariwal (2021) to estimate the expectation over the timestep $s$ during training.

**Joint Sampling.** During inference, we generate samples by starting with pure noise and iteratively denoising it. To accelerate this process, we propose a hybrid sampling procedure that uses different strategies for each data type. For the **continuous** trajectories $\mathbf{Y}$, we employ the deterministic Denoising Diffusion Implicit Model (DDIM) sampler (Song et al., 2020) as in Capellera et al. (2025). It allows for larger jumps in the denoising process. The update rule to go from step $s$ to $s - \zeta$ is:
$\mathbf{Y}_{s-\zeta} = \sqrt{\frac{\bar{\alpha}_{s-\zeta}}{\bar{\alpha}_s}} \mathbf{Y}_s + \left( \sqrt{1 - \bar{\alpha}_{s-\zeta}} - \sqrt{\frac{\bar{\alpha}_{s-\zeta}}{\bar{\alpha}_s}} \sqrt{1 - \bar{\alpha}_s} \right) \epsilon_\theta(\mathbf{Y}_s, \mathbf{E}_s, s, \mathbf{X}^{\text{co}}, \mathcal{G}).$

For the **discrete** events, we use the standard stochastic sampler (Hoogeboom et al., 2021). At each step $s$, the network's classification head predict the original event distribution $\hat{\mathbf{E}}_0 = \pi_\theta(\mathbf{Y}_s, \mathbf{E}_s, s, \mathbf{X}^{\text{co}}, \mathcal{G})$. We then sample $\mathbf{E}_{s-1}$ from the posterior $q(\mathbf{E}_{s-1} \mid \mathbf{E}_s, \hat{\mathbf{E}}_0)$.

**Beta Schedule.** To align the continuous and discrete sampling processes and improve model accuracy, we employ a hybrid schedule where the total number of discrete steps, denoted as $S^d$, is reduced ($S^d < S$). Following Levi et al. (2023), we align the discrete steps ($s^d$) with the continuous ones $s$ using $s^d = \lceil s \cdot (S^d/S) \rceil$. We empirically found that a good choice is to match the DDIM skipping step $\zeta$ with the ratio $S/S^d$. See an empirical evaluation at Appendix D.3.1.

**Controllability.** For controllable generation, we utilize the CFG (Ho & Salimans, 2022) training approach. During training, we randomly drop the condition $\mathcal{G}$ with a probability of 25%, which allows the model to learn to denoise both with and without the conditioning information. For non-controllable generation, the model is trained without any conditioning. During inference, we found that we can achieve effective guidance by using a single forward pass with the conditional output. See an empirical evaluation at Appendix D.3.3.

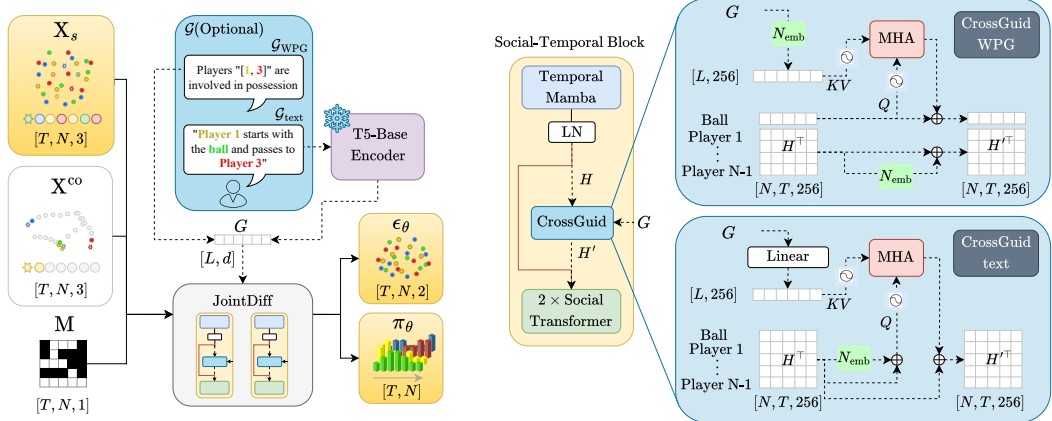

Figure 2: **Model Architecture. Left:** The overall pipeline of our JointDiff model, which takes as input the noisy states $\mathbf{X}_s$, observed states $\mathbf{X}^{\text{co}}$, mask $\mathbf{M}$, and optionally (referred with dashed connections) the encoded guidance signal $G$. Stars (★) refer to the initial timestep, $t = 0$. The model processes these inputs through two Social-Temporal Blocks and outputs the predicted Gaussian noise $\epsilon_\theta$ for trajectories and the event probability distribution $\pi_\theta$. **Right:** Detailed view of a Social-Temporal Block featuring our proposed CrossGuid module. The module has two distinct implementations corresponding to different guidance modalities (WPG and Text). The red line ($-$) in Social-Temporal Block indicates the data flow for non-controllable generation, where CrossGuid is bypassed. An extended diagram is available in Fig. 5.

## 4.3 MODEL ARCHITECTURE

Our model, JointDiff, builds upon the U2Diff architecture (Capellera et al., 2025), which processes multi-agent trajectories using Social-Temporal Blocks. Each block comprises a Temporal Mamba module (Gu & Dao, 2023) for modeling individual agent dynamics and Social Transformers encoders (Vaswani et al., 2017) for capturing inter-agent interactions. We modify this foundation to create a joint diffusion process capable of controllable generation.

As shown in Fig. 2-left, the model takes as input the noisy state $\mathbf{X}_s \in \mathbb{R}^{T \times N \times 3}$, formed by concatenating the continuous trajectory coordinates $\mathbf{Y}_s$ and discrete event indicators $\mathbf{E}_s$ along the feature dimension, and the observed state $\mathbf{X}^{\text{co}} \in \mathbb{R}^{T \times N \times 3}$ (constructed similarly), along with the binary mask $\mathbf{M} \in \mathbb{R}^{T \times N}$. It processes these through two Social-Temporal Blocks and produces two outputs using:

- A regression head that predicts the Gaussian noise $\epsilon_\theta$ for the continuous trajectories.

- A classification head that predicts the probability distribution $\pi_\theta$ for the original discrete events, yielding $\hat{\mathbf{E}}_0$.

To enable controllable generation, we introduce the **CrossGuid** module, which injects an external guidance signal $\mathcal{G}$ into the denoising network. This signal is first encoded into a conditioning tensor $G \in \mathbb{R}^{L \times d}$, where $L$ is the sequence length and $d$ is the feature dimension, both specific to the guidance modality. During training, conditioning dropout ($\mathcal{G} = \emptyset$) is performed by setting $G$ to a zero tensor. Refer to Appendix B for architecture details.

### 4.3.1 CROSSGUID FOR CONTROLLABLE GENERATION

The CrossGuid module is integrated within each Social-Temporal Block, situated between the Temporal Mamba and the first Social Transformer. It refines the intermediate representation $H \in \mathbb{R}^{T \times N \times 256}$ (obtained after processing by the Temporal Mamba and Layer Normalization) using the conditioning tensor $G$. The operation is defined as $H' = \text{CrossGuid}(H, G) = H + \Delta H$, where the residual update $\Delta H$ is computed via a Multi-Head Attention (MHA) mechanism (Vaswani et al., 2017). The implementation varies with the guidance modality, as detailed below and shown in Fig. 2-right, where $H^\top, H'^\top \in \mathbb{R}^{N \times T \times 256}$ are the transposed tensor of $H$ and $H'$, respectively.

**Weak-Possessor-Guidance (WPG).** This modality conditions generation on a sequence of ball possessors. The guidance signal $\mathcal{G}_{\text{WPG}}$ is a sequence $[n_1, n_2, \ldots, n_L]$ where each $n_i \in \{1, \ldots, N-1\}$ denotes a player index. This sequence is encoded as a one-hot matrix $G \in \mathbb{R}^{L \times N}$.

- **Key/Value ($K$, $V$).** Each possessor index in $G$ is mapped through a learnable agent embedding layer $N_{\text{emb}} \in \mathbb{R}^{N \times 256}$, yielding $K = V = N_{\text{emb}}(G) \in \mathbb{R}^{L \times 256}$.

- **Query ($Q$).** The query is derived solely from the ball's intermediate representation: $Q = H[:, 0] \in \mathbb{R}^{T \times 256}$, where index 0 corresponds to the ball.

- **Positional Encoding.** 1D sinusoidal positional encodings (Vaswani et al., 2017) are added to $Q$ (along the temporal dimension $T$) and to $K$, $V$ (along the possession sequence dimension $L$).

- **MHA and Update.** The update is applied only to the ball's trajectory: $H'[:, 0] = H[:, 0] + \text{MHA}(Q, K, V)$.

- **Agent Embedding Addition.** To facilitate social reasoning, the learnable agent embedding for each player $n \in \{1, \ldots, N-1\}$ is added to their respective representation: $H'[:, n] = H[:, n] + N_{\text{emb}}(n)$.

**Text-Guidance.** This modality conditions generation on a natural language prompt $\mathcal{G}_{\text{text}}$. The text is tokenized and encoded using a frozen, pre-trained T5-Base Encoder (Raffel et al., 2020), producing $G \in \mathbb{R}^{L \times 768}$, where $L$ is the number of tokens and $d = 768$.

- **Key/Value ($K$, $V$).** The text embeddings are projected to the model's dimension: $K = V = \text{Linear}(G) \in \mathbb{R}^{L \times 256}$.

- **Query ($Q$).** The query is formed from the representation of all agents. To distinguish between agents, the learnable agent embedding is added before projection: $Q[:, n] = H[:, n] + N_{\text{emb}}(n) \in \mathbb{R}^{T \times 256}$ for each agent $n$.

- **Positional Encoding.** 1D positional encodings are added to each agent's query $Q[:, n]$ (along time) and to $K$, $V$ (along the text token sequence).

- **MHA and Update.** The MHA operation is performed independently for each agent against the shared textual context. The update is applied to all agents: $H'[:, n] = H[:, n] + \text{MHA}(Q[:, n], K, V)$ for $n \in \{0, \ldots, N-1\}$.

## 5 EXPERIMENTS

**Continuous trajectories (Y).** We validate JointDiff on three public sports datasets: **NBA**, **NFL**, and **Bundesliga**. The NBA dataset uses the widely adopted SportVU data[1], with the splits from Mao et al. (2023) (32.5k training / 12k testing scenes). Each scene spans 6 seconds ($T = 30$ timesteps, 5 fps) with $N = 11$ agents (the ball and 10 players). The NFL dataset comes from the Big Data Bowl[2], following the splits of Xu & Fu (2025) (10,762 training / 2,624 testing scenes). Each scene covers 5 seconds ($T = 50$, 10 fps) with $N = 23$ agents (the ball and 22 players). Finally, the Bundesliga dataset is curated from the German soccer league[3] (Bassek et al., 2025), containing 2,093 training and 524 testing scenes from 7 matches. Scenes with fewer than $N = 23$ agents or out-of-play were removed. Each scene spans 6.4 seconds ($T = 40$ timesteps, 6.25 fps) with $N = 23$ agents, and the training set is augmented with 180° rotations, doubling its size.

**Possessor event (E).** To compare with methods that do not model events, we extract possessor events from trajectories **Y** using a simple heuristic: a player possesses the ball if it is within 1.5 meters (see Appendix C.1). When multiple players are in range, the closest is chosen; if none, we assign the ball as the possessor, acting as no possessor class (e.g., during pass or shot).

**Guidance data ($\mathcal{G}$).** From the possessor events, we generate the weak-possessor-guidance signal ($\mathcal{G}_{\text{WPG}}$), a sequence of unique consecutive players filtered from the ground-truth events (e.g., $\mathbf{E} = [1, 1, 1, 1, 0, 0, 0, 0, 3, 3, 3]$ yields $\mathcal{G}_{\text{WPG}} = [1, 3]$). We also create natural language descriptions ($\mathcal{G}_{\text{text}}$) for NFL and Bundesliga using public metadata. NFL events are aligned with tracking data via

---

[1]https://github.com/linouk23/NBA-Player-Movements
[2]https://github.com/nfl-football-ops/Big-Data-Bowl
[3]https://github.com/spoho-datascience/idsse-data

Table 1: **Completion Generation.** The table reports results for our JointDiff and state-of-the-art baselines, solving the completion generation task. We report performance metrics for two distinct tasks: Future Generation (**top**) and Imputation Generation (**bottom**). Performance metrics computed over 20 generated modes, using $min$ / $avg$, with the exception of the uni-modal method, which are noted as having 1 mode in the IID column. The Gen column specifies whether a model is generative.

| Method | Gen | IID | NFL (yards) | | Bundesliga (meters) | | NBA (meters) | |
|---|---|---|---|---|---|---|---|---|
| | | | SADE↓ | SFDE↓ | SADE↓ | SFDE↓ | SADE↓ | SFDE↓ |
| GroupNet CVPR22 | ✓ | ✓ | 4.42 / 5.33 | 10.01 / 12.18 | 4.78 / 5.76 | 9.58 / 11.63 | 2.12 / 2.84 | 3.72 / 5.15 |
| AutoBots ICLR22 | ✗ | ✗ | 3.02 / 4.82 | 6.33 / 10.68 | 3.33 / 5.93 | 5.57 / 11.46 | 1.75 / 2.73 | 2.73 / 4.71 |
| LED[IID] CVPR23 | ✓ | ✓ | 3.48 / 4.12 | 7.95 / 9.63 | 3.89 / 4.58 | 8.06 / 9.74 | 1.77 / 2.30 | 3.25 / 4.45 |
| LED CVPR23 | ✓ | ✗ | - | - | - | - | 1.63 / 3.83 | 2.99 / 6.03 |
| MART ECCV24 | ✗ | ✗ | 2.55 / 4.26 | 5.99 / 10.31 | 2.50 / 4.16 | 5.06 / 9.00 | 1.52 / 2.46 | 2.77 / 4.78 |
| MoFlow CVPR25 | ✓ | ✗ | **2.33** / 4.02 | **5.51** / 9.98 | 2.51 / 4.21 | 5.08 / 9.24 | 1.52 / 2.42 | 2.73 / 4.64 |
| U2Diff CVPR25 | ✓ | ✓ | 2.59 / 3.74 | 5.97 / 9.02 | 2.69 / 4.21 | 5.46 / 9.44 | 1.48 / 2.12 | 2.68 / 4.14 |
| JointDiff (Ours) | ✓ | ✓ | 2.36 / **3.40** | 5.53 / **8.40** | **2.47** / **3.66** | **5.02** / **8.29** | **1.39** / **2.01** | **2.53** / **3.95** |
| TranSPORTmer ACCV24 | ✗ | 1 | 1.27 | - | 1.45 | - | 0.71 | - |
| Sports-Traj ICLR25 | ✓ | ✓ | 2.28 / 2.29 | - | 2.75 / 2.75 | - | 1.19 / 1.20 | - |
| U2Diff CVPR25 | ✓ | ✓ | 0.96 / 1.19 | - | 1.04 / 1.36 | - | 0.62 / 0.83 | - |
| JointDiff (Ours) | ✓ | ✓ | **0.84** / **1.03** | - | **0.91** / **1.18** | - | **0.57** / **0.78** | - |

Table 2: **Controllable Generation.** The table reports results for our JointDiff and a variant without our joint framework (Ours w/o joint), solving the controllable future generation task. It is also included performance on the non-controllable task (w/o $\mathcal{G}$) as well as two controllable tasks: WPG (w $\mathcal{G}_{\text{WPG}}$) and text-guidance (w $\mathcal{G}_{\text{text}}$). Performance metrics computed over 20 generated modes, using $min$ / $avg$ for SADE and SFDE and $max$ / $avg$ for Acc.

| Method | NFL (yards) | | | Bundesliga (meters) | | | NBA (meters) | | |
|---|---|---|---|---|---|---|---|---|---|
| | SADE ↓ | SFDE ↓ | Acc ↑ | SADE ↓ | SFDE ↓ | Acc ↑ | SADE ↓ | SFDE ↓ | Acc ↑ |
| Ours w/o joint | | | | | | | | | |
| w/o $\mathcal{G}$ | 2.42 / 3.57 | 5.67 / 8.72 | .76 / .52 | 2.60 / 3.99 | 5.30 / 8.95 | .67 / .44 | 1.46 / 2.13 | 2.64 / 4.19 | .74 / .44 |
| w $\mathcal{G}_{\text{WPG}}$ | 2.37 / 3.49 | 5.51 / 8.49 | .80 / .59 | 2.20 / 3.07 | 4.35 / 6.71 | .73 / .50 | 1.29 / 1.91 | 2.27 / 3.74 | .86 / .66 |
| w $\mathcal{G}_{\text{text}}$ | 2.33 / 3.39 | 5.40 / 8.25 | .80 / .63 | 2.16 / 2.96 | 4.18 / 6.15 | .78 / .55 | - | - | - |
| Ours | | | | | | | | | |
| w/o $\mathcal{G}$ | 2.36 / 3.40 | 5.53 / 8.40 | .78 / .54 | 2.47 / 3.66 | 5.02 / 8.29 | .68 / .39 | 1.39 / 2.01 | 2.53 / 3.95 | .75 / .45 |
| w $\mathcal{G}_{\text{WPG}}$ | 2.29 / 3.26 | 5.29 / 7.94 | .84 / .65 | 2.13 / 2.85 | 4.22 / 6.16 | .77 / .52 | 1.24 / 1.81 | 2.20 / 3.53 | .87 / .67 |
| w $\mathcal{G}_{\text{text}}$ | 2.19 / 3.09 | 5.04 / 7.52 | .86 / .74 | 2.08 / 2.72 | 4.09 / 5.68 | .80 / .59 | - | - | - |

possessor information, while Bundesliga follows Kim et al. (2025). This fine-grained conditioning provides more control than $\mathcal{G}_{\text{WPG}}$. Refer to Appendix C.2 for more details. The code to generate the datasets and the guidance data will be released jointly with unified dataloader to constitute an easy-usable benchmark for future works.

**Implementation.** We use $S = 50$ diffusion steps for continuous and $S^d = 10$ for discrete data, with a shared quadratic noise scheduler from $\beta_0 = 10^{-4}$ to $\beta_S = 0.5$ (Capellera et al., 2025; Tashiro et al., 2021). The discrete loss coefficient is set to $\lambda = 0.1$, which provided the best trade-off between trajectory and event accuracy (see Appendix D.3.2). Sampling employs DDIM with a skip interval $\zeta = 5$ and an extra denoising step at $s = 1$, yielding 11 steps: $\{50, 45, \ldots, 5, 1, 0\}$. Training runs for 100 epochs (NBA/NFL) and 200 epochs (Bundesliga) with batch size 16; learning rate $10^{-3}$ is halved every 20 (NBA/NFL) or 40 (Bundesliga) epochs. The model uses a hidden size of 256 and 8 attention heads in all multi-head attention layers, while the Social Transformer employs a 1024-dimensional feedforward layer. All models are trained on a single RTX A6000.

## 5.1 COMPLETION GENERATION

We evaluate completion on two sub-tasks: future generation and imputation generation. For each scene, we sample 20 modes and report both minimum ($min$) and average ($avg$) errors, where $min$ reflects the best-case generation quality and $avg$ measures distributional fidelity.

In **future generation**, models observe 10 frames and predict the future. We report scene-level errors as SADE and SFDE (Casas et al., 2020; Girgis et al., 2021; Weng et al., 2023; Capellera et al., 2025). Prior work can be grouped by how multiple futures are produced. IID models (GroupNet

(Xu et al., 2022), LED^IID (Mao et al., 2023), U2Diff (Capellera et al., 2025), ours) draw modes independently from random noise. As image generation, this encourages sampling-fidelity with the real data distribution and usually yields stronger $avg$ performance. In contrast, non-IID models (AutoBots (Girgis et al., 2021), LED (Mao et al., 2023), MART (Lee et al., 2024), MoFlow (Fu et al., 2025)) generate multiple correlated modes in a single forward pass, which often improves $min$ metrics empirically. As shown in Table 1-top, JointDiff achieves SOTA across datasets in $avg$, while, notably competing with $min$ metrics against non-IID approaches. We also note whether a model is generative or not. Refer to Appendix D.1 for the implementation details of these baselines.

In the **imputation generation** setting, models are provided with the first 10 frames and the final frame, and must predict the missing in-between trajectories. We evaluate against the IID models Sports-Traj (Xu & Fu, 2025), U2Diff, and the deterministic uni-modal TranSPORTmer (Capellera et al., 2024), reporting the SADE metric. As shown in Table 1-bottom, our method achieves the SOTA in this benchmark. The results show that diffusion-based models, such as U2Diff (Capellera et al., 2025) and Ours, consistently outperform the CVAE-based model Sports-Traj (Xu & Fu, 2025), which suffers from mode collapse. Additional quantitative and qualitative results are provided in Appendix D.2 and D.5.1, respectively.

To assess perceptual quality, we conduct a **human evaluation** using pairwise comparisons on the NBA **future generation** task. Fifteen participants judged fifteen random pairs drawn from JointDiff, our ablated variant without joint modeling ("w/o joint"), U2Diff, MoFlow, and Ground Truth (GT) (interface shown in Appendix D.2.1). For fairness, each model generated 20 modes from the same past, and the sample with the lowest SADE was used. Results in Fig. 3 show that JointDiff is most preferred, outperforming MoFlow (80%), U2Diff (65%), and w/o joint (53%). Removing ties increases the win rate over the ablated variant to 67%, indicating clear perceptual gains from joint modeling. JointDiff loses to GT in 44% of comparisons, with 24% ties, suggesting that many generated trajectories are difficult for users to distinguish from real ones.

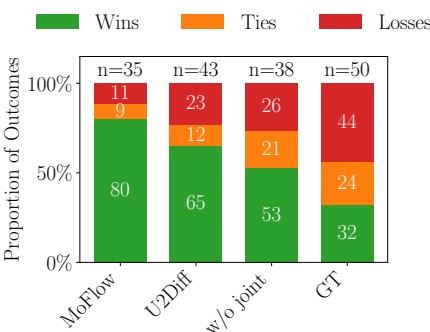

Figure 3: **Human evaluation** on NBA future generation. The histogram reports the proportions of wins, ties, and losses for JointDiff against each baseline, with $n$ denoting the number of pairwise comparisons.

### 5.2 Controllable Generation

We evaluate controllability by generating 20 future trajectories conditioned on ground-truth guidance signals ($\mathcal{G}$). For the trajectories ($\mathbf{Y}_0$), we report the ($min$ / $avg$) for SADE and SFDE. To assess the possessor prediction, we report accuracy (Acc) as the match between $\arg\max(\hat{\mathbf{E}}_0)$ and the ground truth $\mathbf{E}_0$, giving both maximum and average ($max$ / $avg$) over the generated modes.

Table 2 compares non-controlled and controlled tasks and evaluates our joint modeling approach against a variant modeling only continuous trajectories ("Ours w/o joint"). Controlled tasks consistently outperform their non-controlled counterparts ("w/o $\mathcal{G}$"), confirming the effectiveness of CrossGuid. Conditioning on textual signals $\mathcal{G}_{\text{text}}$ further improves performance over $\mathcal{G}_{\text{WPG}}$, showing the benefit of fine-grained guidance. JointDiff surpasses Ours w/o joint overall across all datasets and metrics, validating the advantage of jointly modeling continuous trajectories and discrete possession events and improving denoising in both controlled and non-controlled settings. Figure 4 provides qualitative comparisons using a Bundesliga sample for the same past observations $\mathbf{X}^{\text{co}}$ conditioned on user-defined text prompts $\mathcal{G}_{\text{text}}$, showing improved controllable accuracy. See Appendix for additional ablations (D.3), interpretability analysis (D.4) and qualitative examples (D.5.2). Please refer to the video supplementary to see animated results.

### 5.3 Consistency Analysis

We evaluate the consistency of the joint generation $\hat{\mathbf{X}}_0 = (\hat{\mathbf{Y}}_0, \hat{\mathbf{E}}_0)$. Using the same setup as in Table 2, we generate 20 samples per scene and compute the Acc metric between the predicted discrete possessor $\hat{\mathbf{E}}_0$ and the threshold-based heuristic possessor extracted from the predicted trajectories

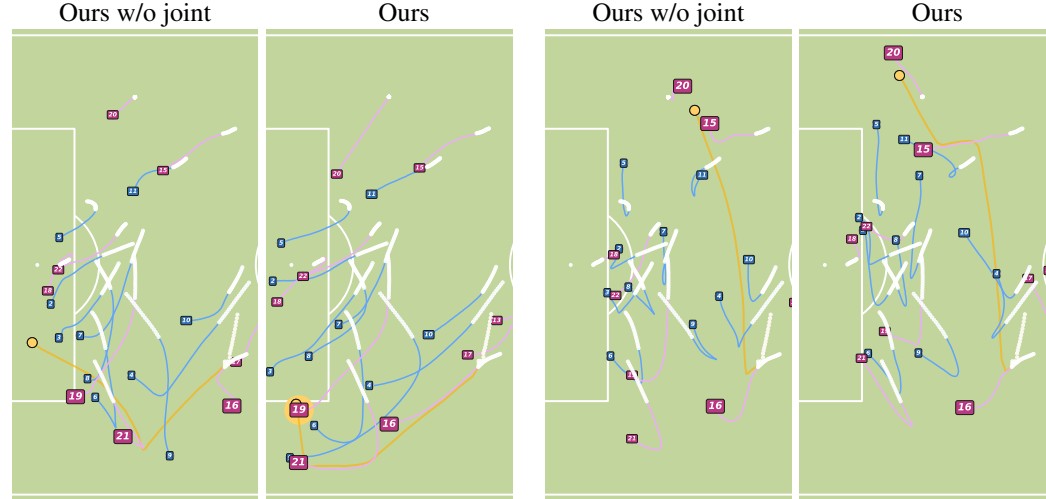

(a) "Away Team has the **possession**. The ball starts at left-center without a carrier. Player 16 **possesses** the ball, moving it from left-center to down-side, then **passes** to Player 21 who **possesses** the ball. Then Player 21 makes a **pass** to Player 19 into the box."

(b) "Away Team has the **possession**. The ball starts at left-center without a carrier. Player 16 makes a **pass** to Player 15 who **receives**. Player 15 **possesses** the ball in left-center and **pass** up-side to Player 20."

Figure 4: **Controllable Generation.** Comparison of JointDiff vs. Ours w/o joint on the text-guidance task giving the same past observations with different text prompts $\mathcal{G}_{\text{text}}$. Legend: 🟡 Ball, 🟦 Home team, 🟥 Away team, ⚪ Past observations. **See animated scenes in supplementary**.

Table 3: **Consistency Analysis.** Consistency between predicted events and trajectories, reported as $max$ / $avg$ Acc $\uparrow$ over 20 samples. We compare our multinomial with the absorbing state framework.

| Method | Multinomial (Ours) | | | Absorbing | | |
|---|---|---|---|---|---|---|
| | NFL | Bundesliga | NBA | NFL | Bundesliga | NBA |
| Ours w/o $\mathcal{G}$ | **.98** / **.86** | **.97** / **.80** | **.99** / **.92** | .97 / .80 | .94 / .70 | **.99** / .89 |
| Ours w $\mathcal{G}_{\text{WPG}}$ | .96 / .84 | .95 / .80 | **.99** / **.92** | .93 / .78 | .91 / .68 | **.99** / .90 |
| Ours w $\mathcal{G}_{\text{text}}$ | .97 / .86 | **.96** / **.81** | - | .95 / .81 | .95 / .76 | - |

$\hat{\mathbf{Y}}_0$. For reference, we compare our multinomial diffusion with the commonly used absorbing state formulation (Levi et al., 2023; Li et al., 2025). For the absorbing, we use the publicly available implementation from Li et al. (2025) and set the absorption rate to $\lambda = 0.01$ to match the magnitude of the discrete loss with the continuous one. Unlike the multinomial model, which enables refinement through all states at each denoising step, the absorbing mechanism freezes tokens once unmasked, preventing later correction (von Rütte et al., 2025). As shown in Table 3, our method achieves high consistency, particularly for the best sample ($max$), and strong consistency on average ($avg$). The larger variance in NFL and Bundesliga aligns with their smaller training sets. Overall, this ablation shows that multinomial diffusion yields substantially more consistent predictions than the absorbing state approach in our dynamic domain. To assess the quality of the generations using absorbing state with respect to the ground truth, results for future generation are shown in Appendix D.3.5.

## 6 CONCLUSIONS

We have introduced JointDiff, a novel diffusion framework that unifies the generation of continuous trajectories and synchronous discrete events in multi-agent systems. Our model obtains state-of-the-art results on completion tasks and enables new forms of semantic control through WPG and text-guidance. We demonstrate that the joint formulation is a key factor, as it significantly enhances the fidelity of the controllable generation. While this work assumes a dense event pattern, a promising future direction is to extend the framework to sparse events, such as time point processes. Finally, JointDiff provides a strong foundation for generating controllable, low-dimensional data to steer high-dimensional generative models, such as for video synthesis in complex interactive domains.

**Acknowledgment.** This work has been supported by the project GRAVATAR PID2023-151184OB-I00 funded by MCIU/AEI/10.13039/501100011033 and by ERDF, UE and by the Gov. of Catalonia under 2023 DI 00058.

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

## A  DIFFUSION BACKGROUND

This section is intended to be an extension of the diffusion background provided in the main paper (Section 3). We rescue the notation from the main paper for clarity and detail the specific processes for continuous and discrete data.

### A.1  CONTINUOUS CASE

The data is corrupted with Gaussian noise until it converges to a standard isotropic Gaussian distribution. The forward process at any step $s$ can be stated in both recursive and closed form as:

$$q(\mathbf{X}_s \mid \mathbf{X}_{s-1}) = \mathcal{N}(\mathbf{X}_s; \sqrt{1-\beta_s}\mathbf{X}_{s-1}, \beta_s \mathbf{I}), \tag{9}$$

$$q(\mathbf{X}_s \mid \mathbf{X}_0) = \mathcal{N}(\mathbf{X}_s; \sqrt{\bar{\alpha}_s}\mathbf{X}_0, (1-\bar{\alpha}_s)\mathbf{I}). \tag{10}$$

This closed-form expression allows us to sample $\mathbf{X}_s$ directly from $\mathbf{X}_0$ using the reparameterization trick:

$$\mathbf{X}_s = \sqrt{\bar{\alpha}_s}\mathbf{X}_0 + \sqrt{1-\bar{\alpha}_s}\epsilon, \quad \text{where } \epsilon \sim \mathcal{N}(\mathbf{0}, \mathbf{I}). \tag{11}$$

The reverse process generates data by starting with a sample from the prior, $\mathbf{X}_S \sim \mathcal{N}(\mathbf{0}, \mathbf{I})$, and iteratively sampling from the reverse conditional distributions $p_\theta(\mathbf{X}_{s-1} \mid \mathbf{X}_s)$. The true posterior $q(\mathbf{X}_{s-1} \mid \mathbf{X}_s)$ is intractable as it depends on the entire data distribution. Therefore, we approximate it with a neural network $p_\theta(\cdot)$ that outputs the parameters of a Gaussian:

$$p_\theta(\mathbf{X}_{s-1} \mid \mathbf{X}_s) = \mathcal{N}(\mathbf{X}_{s-1}; \boldsymbol{\mu}_\theta(\mathbf{X}_s, s), \sigma_s^2 \mathbf{I}). \tag{12}$$

The mean $\boldsymbol{\mu}_\theta$ is commonly parameterized in terms of a predicted noise term $\epsilon_\theta(\mathbf{X}_s, s)$:

$$\boldsymbol{\mu}_\theta(\mathbf{X}_s, s) = \frac{1}{\sqrt{\alpha_s}}\left(\mathbf{X}_s - \frac{\beta_s}{\sqrt{1-\bar{\alpha}_s}}\epsilon_\theta(\mathbf{X}_s, s)\right). \tag{13}$$

For simplicity, the variance $\sigma_s^2$ is typically set to a non-learned constant, such as $\sigma_s^2 = \beta_s$. With this parameterization, minimizing the variational bound $\mathcal{L}_{\mathrm{vb}}$ (Eq. 1) is equivalent to training the noise prediction network $\epsilon_\theta$ with a simplified objective:

$$\mathcal{L}_{\mathrm{simple}} = \mathbb{E}_{s, \mathbf{X}_0, \epsilon}\left[\|\epsilon - \epsilon_\theta(\sqrt{\bar{\alpha}_s}\mathbf{X}_0 + \sqrt{1-\bar{\alpha}_s}\epsilon, s)\|_2^2\right]. \tag{14}$$

### A.2  DISCRETE CASE

Here, the data is categorical and typically represented by one-hot vectors of dimension $N$. The forward process corrupts the data until it converges to a uniform distribution across all $N$ categories. This is defined as a multinomial diffusion process:

$$q(\mathbf{X}_s \mid \mathbf{X}_{s-1}) = \mathrm{Cat}(\mathbf{X}_s; (1-\beta_s)\mathbf{X}_{s-1} + \beta_s/N), \tag{15}$$

$$q(\mathbf{X}_s \mid \mathbf{X}_0) = \mathrm{Cat}(\mathbf{X}_s; \bar{\alpha}_s\mathbf{X}_0 + (1-\bar{\alpha}_s)/N). \tag{16}$$

The posterior is a categorical distribution $q(\mathbf{X}_{s-1} \mid \mathbf{X}_s, \mathbf{X}_0) = \mathrm{Cat}(\mathbf{X}_{s-1}; , \boldsymbol{\theta}_{\mathrm{post}}(\mathbf{X}_s, \mathbf{X}_0))$ whose probabilities are defined as:

$$\boldsymbol{\theta}_{\mathrm{post}}(\mathbf{X}_s, \mathbf{X}_0) = \tilde{\boldsymbol{\theta}}/\sum_n \tilde{\boldsymbol{\theta}}_n \quad \text{and} \quad \tilde{\boldsymbol{\theta}} = [\alpha_s \mathbf{X}_s + (1-\alpha_s)/N] \odot [\bar{\alpha}_{s-1}\mathbf{X}_0 + (1-\bar{\alpha}_{s-1})/N].$$

Here, $\odot$ denotes the element-wise product. Note that the result is normalized to sum to one.

The reverse process learns to approximate the true posterior as in Eq. 1. It does so by learning to approximate the original clean data $\mathbf{X}_0$ from a noisy version $\mathbf{X}_s$. Indeed, a probability vector is predicted using a neural network $\pi_\theta(\mathbf{X}_s, s) = \hat{\mathbf{X}}_0$. Subsequently, we can parametrize when $s = 1$ and $s \geq 2$ respectively:

$$p_\theta(\mathbf{X}_0 \mid \mathbf{X}_1) = \mathrm{Cat}(\mathbf{X}_0; \hat{\mathbf{X}}_0) \quad \text{and} \quad p_\theta(\mathbf{X}_{s-1} \mid \mathbf{X}_s) = q(\mathbf{X}_{s-1} \mid \mathbf{X}_s, \hat{\mathbf{X}}_0). \tag{17}$$

The $D_{\mathrm{KL}}$ expressions for $s \geq 2$ in Eq. 1 can be computed as:

$$D_{\mathrm{KL}}(q(\mathbf{X}_{s-1} \mid \mathbf{X}_s, \mathbf{X}_0) \,\|\, p_\theta(\mathbf{X}_{s-1} \mid \mathbf{X}_s)) = D_{\mathrm{KL}}\Big(\boldsymbol{\theta}_{\mathrm{post}}(\mathbf{X}_s, \mathbf{X}_0) \,\|\, \boldsymbol{\theta}_{\mathrm{post}}(\mathbf{X}_s, \hat{\mathbf{X}}_0))\Big), \tag{18}$$

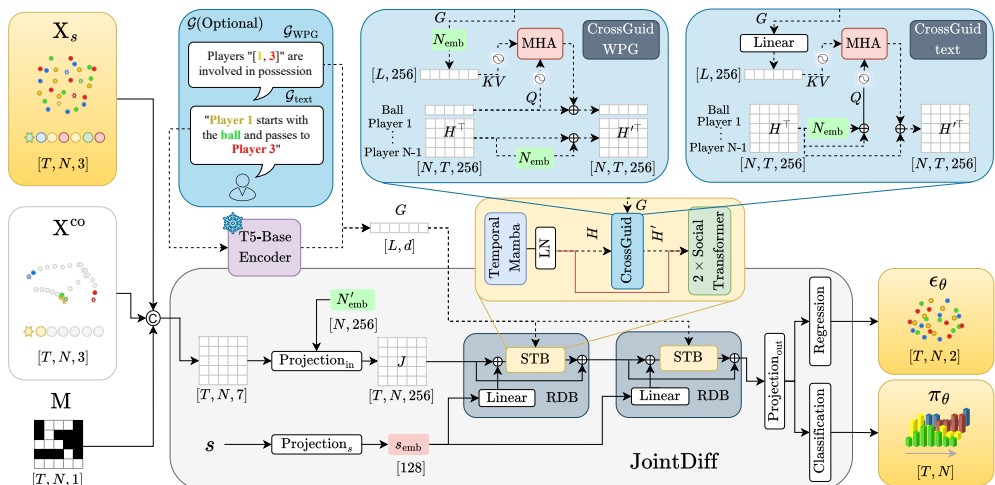

Figure 5: **Model Architecture (extended version of Fig. 2).** The light gray box represents the JointDiff architecture, and STB refers to the Social-Temporal Block.

by enumerating the probabilities and using $\sum_n \boldsymbol{\theta}_{\text{post}}(\mathbf{X}_s, \mathbf{X}_0)_n \cdot \log \frac{\boldsymbol{\theta}_{\text{post}}(\mathbf{X}_s, \mathbf{X}_0)_n}{\boldsymbol{\theta}_{\text{post}}(\mathbf{X}_s, \hat{\mathbf{X}}_0)_n}$. The log-likelihood term can be computed as:

$$p_\theta(\mathbf{X}_0 \mid \mathbf{X}_1) = \sum_n \mathbf{X}_{0,n} \cdot \log \hat{\mathbf{X}}_{0,n}. \tag{19}$$

## B  ARCHITECTURE

We provide a detailed explanation of the general architecture, which can be shown in Fig.5.

The model takes as input the noisy states $\mathbf{X}_s$, observed states $\mathbf{X}^{\text{co}}$ with their corresponding binary mask $\mathbf{M}$, the denoising step $s$, and the optional guidance signal $\mathcal{G}$. The noisy state $\mathbf{X}_s$ is formed by concatenating the continuous trajectories $\mathbf{Y}_s$ with the discrete possession events $\mathbf{E}_s$. The observed states $\mathbf{X}^{\text{co}}$ are similarly constructed using $\mathbf{Y}^{\text{co}}$ and $\mathbf{E}^{\text{co}}$. These two $[T, N, 3]$ tensors, along with the binary mask $\mathbf{M}$, are then concatenated to create a single input tensor of dimension $[T, N, 7]$. This input tensor is first processed through a layer (Projection$_{\text{in}}$) which embeds the data and combines it with a learnable agent embedding ($N'_{\text{emb}}$). This results in a embedding tensor $J$ of dimension $[T, N, 256]$. The denoising step $s$ is embedded through a layer (Projection$_s$) to become an embedding $s_{\text{emb}} \in \mathbb{R}^{128}$. Both $J$ and $s_{\text{emb}}$ are then processed by a sequence of two residual denoising blocks (RDBs). Within each RDB, the $s_{\text{emb}}$ is added to $J$ after a Linear embedding projection to dimension 256. Then the resulting tensor is processed by a Social-Temporal Block (STB). As explained in the main paper, for controllable generation, we introduce an operation called **CrossGuid** inside this block to guide the denoising process with the signal $\mathcal{G}$. After the RDBs processing, the resulting embedding with the same dimension as $J$ is processed through a layer (Projection$_{\text{out}}$), defining an output tensor embedding which is then projected in into two heads: a regression head (Regression), based on a linear layer, to predict the Gaussian noise $\epsilon_\theta$ added to the continuous trajectories; and a classification head (Classification), based on a linear layer followed by a softmax activation, to predict the probability distribution of the original discrete events, yielding $\hat{\mathbf{E}}_0 = \pi_\theta$.

Now we define each mentioned operation:

- **Projection$_{\text{in}}$**: The input tensor of shape $[T, N, 7]$ is combined with a learnable agent embedding $N'_{\text{emb}} \in \mathbb{R}^{[N, 256]}$. First, the input tensor is linearly projected to $[T, N, 256]$ and concatenated with the agent embeddings (replicated $T$ times along the temporal axis), yielding a $[T, N, 512]$ tensor. A Linear layer with ReLU activation re-projects this tensor to $[T, N, 256]$, producing the embedding tensor $J$.

- **Projection$_s$**: Embeds the denoising timestep $s$ using a Linear layer followed by a SiLU activation, producing $s_{\text{emb}} \in \mathbb{R}^{128}$.

- **RDB**: Processes $J$ conditioned on $s_{\text{emb}}$, and optionally the guidance embedding $G$, as:
$$\text{RDB}(J, s_{\text{emb}}, G) = J + \text{STB}(J + \text{Linear}(s_{\text{emb}}), G),$$

- **Social-Temporal Block (STB)**: Maps $[T, N, 256] \mapsto [T, N, 256]$ via
$$\text{STB}(J, G) = \text{ST}\big(\text{ST}\big(\text{CrossGuid}(\text{LN}(\text{TM}(J)), G)\big)\big),$$

  where TM denotes the Temporal Mamba, LN is Layer Normalization, CrossGuid is the proposed guidance module, and ST is the Social Transformer. For non-controllable tasks, CrossGuid reduces to the identity.

- **Temporal Mamba (TM)**: As described in Capellera et al. (2025), TM applies bidirectional Mamba modules (Gu & Dao, 2023) independently to each agent and sums the results.

- **Social Transformer (ST)**: A Transformer encoder (Vaswani et al., 2017) applied per timestep to model inter-agent correlations.

- **Projection$_{\text{out}}$**: A Linear layer with ReLU maps $[T, N, 256] \mapsto [T, N, 256]$.

- **Regression**: A Linear layer maps $[T, N, 256] \mapsto [T, N, 2]$, producing the predicted noise $\epsilon_\theta$.

- **Classification**: A Linear layer followed by a Softmax along the agent axis maps $[T, N, 256] \mapsto [T, N]$, producing possessor probabilities $\pi_\theta$.

## C  DATASETS

### C.1  POSSESSOR THRESHOLD

To robustly define the discrete possessor event data ($\mathbf{E}$) from the continuous trajectories ($\mathbf{Y}$), we conducted a data-driven analysis to determine an optimal geometric threshold. Our goal was to identify the distance at which a player is most likely to be in possession of the ball.

Our methodology is based on the key observation that a ball's trajectory is primarily linear when it is not in any player's possession (e.g., during a pass or a shot). Conversely, a player taking possession or influencing the ball's path will induce a significant change in its direction. To quantify this, we analyzed the change in the ball's direction by computing the angle between consecutive ball velocity vectors across our training datasets. For a range of distance thresholds from 0 to 3 meters, we calculated the average angle of change for the ball only during periods when it was outside of that specific threshold distance from all players. The results are shown in Fig. 6.

The optimal threshold was defined as the minimum distance that minimizes this average angle of change. This approach allowed us to identify the point at which the ball's trajectory becomes most linear, indicating the absence of player control. Our analysis across all sports consistently identified approximately **1.5 meters** as the optimal threshold, supporting the use of a single, unified geometric heuristic for defining possession events across different sports. This finding aligns with the observation that player influence on the ball's trajectory is negligible beyond this distance.

### C.2  TEXTUAL GUIDANCE DATASET GENERATION

The core methodology for caption generation is consistent across both Bundesliga and NFL datasets, ensuring a standardized approach to data creation. The pipeline consists of the following stages:

**Stage 1: Preprocessing and Standardization**  The process begins with raw spatio-temporal tracking data, which is first segmented into fixed-length sequences. To ensure data consistency and privacy, all entities are anonymized. Players are systematically numbered from 1 to 22 (with home players assigned 1–11 and away players 12–22), aligning their identifiers with their corresponding order in the final trajectory tensors.

**Stage 2: Automated Feature Extraction**  For each standardized sequence, a script programmatically analyzes the trajectory and events data to extract key semantic features. This automated analysis includes identifying the ball possessor at each frame, detecting discrete

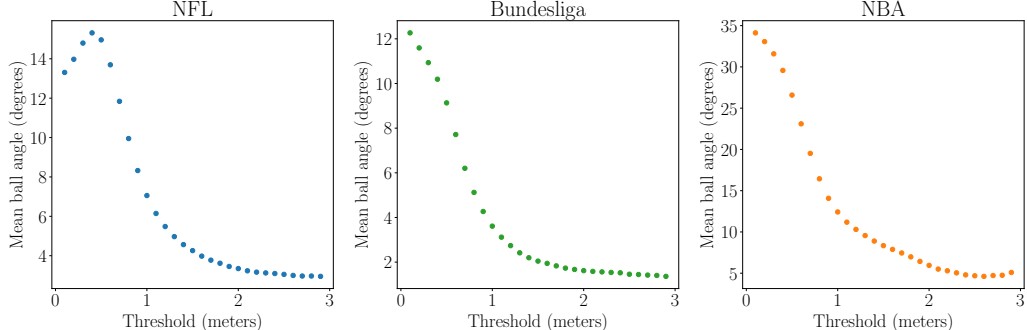

Figure 6: **Possession threshold determination.** Average ball direction change versus distance threshold. The minimum at 1.5 meters indicates optimal threshold where ball motion is most linear without player influence, supporting our unified possession detection heuristic.

game events (e.g., passes, tackles), and mapping the ball's location to meaningful zones on the field. These features are then compiled into a structured, rule-based textual summary, which we term a *dense caption*.

**Stage 3: LLM-based Narrative Refinement** In the final stage, the dense captions are transformed into fluent, human-readable narratives. We leverage a Large Language Model (LLM), prompting it to act as an expert sports analyst. The LLM uses the structured information from the dense caption to generate a chronologically accurate and natural-sounding description that adheres to the common parlance of the respective sport.

While the pipeline is shared, its parameters and feature extraction rules are tailored to the unique characteristics of each sport. Refer to Fig.13 to see examples of these two datasets.

### C.2.1 NFL DATASET

- **Data Source:** We process raw tracking data from the NFL Big Data Bowl, adopting the data splits and experimental settings from Xu & Fu (2025).
- **Sequence Processing:** Trajectories are segmented into 50-frame sequences sampled at 10 fps (5.0 seconds in duration).
- **Feature Extraction:**
  - *Possession:* The player closest to the ball within a 1.5-meter threshold is designated the ball carrier.
  - *Events:* Key extracted events include `ball_snap`, `pass_forward`, and `tackle`.
  - *Formation:* Offensive team spatial configuration.
  - *Short textual description:* A natural language description of the final outcome of the play, covering a time horizon of more than one split
  - *Location:* The ball's position is mapped to the corresponding yard line.

### C.2.2 BUNDESLIGA DATASET

- **Data Source:** We use the integrated trajectory and event dataset provided by Bassek et al. (2025).
- **Sequence Processing:** Raw 25 fps tracking data is synchronized with asynchronous event data, downsampled to 6.25 fps, and segmented into 40-frame sequences (6.4 seconds in duration). Sequences with less than 23 agents and out-of-play are discarded.
- **Feature Extraction:**
  - *Possession:* The player closest to the ball within a 1.5-meter threshold is identified as the possessor.
  - *Events:* Key extracted events include `pass`, `tackle`, and `shot`.

– *Location:* The ball's coordinates are mapped to a predefined semantic grid that partitions the field into named zones (e.g., `down-corner`, `box`).

# D EXPERIMENTS

## D.1 BASELINES

To construct the Table 1 we implemented and evaluated several state-of-the-art architectures. Below we detail how results were obtained for each method:

- **GroupNet** (Xu et al., 2022): We used the official code and checkpoints for NBA. For NFL and Bundesliga, we adapted the parameter `hyper_scales` to $[11, 23]$ to match the number of agents, and doubled both `hidden_dim` and `zdim`.

- **AutoBots** (Girgis et al., 2021): We used the official repository with the same hyperparameters as in TrajNet++. The AutoBotJoint variant was trained on all three datasets.

- **LED$^{\text{IID}}$** (Mao et al., 2023): We used the official code and checkpoints for NBA. For NFL and Bundesliga, we adapted the code to handle longer horizons and adjusted hyperparameters. Following the users' recommended procedure, we first pre-trained the denoiser on a single timestep prediction task, then fine-tuned it for full temporal horizon prediction. Both trainings were performed using 100 epochs, batch size equal to 250, and a learning rate of $10^{-3}$.

- **LED** (Mao et al., 2023): Official code and checkpoints were used for NBA. We were unable to train this stage for NFL and Bundesliga due to GPU memory limitations.

- **MART** (Lee et al., 2024), **MoFlow** (Fu et al., 2025): Official code and checkpoints were used for NBA. For NFL and Bundesliga, we trained using the same settings, changing only the prediction horizon.

- **Sports-Traj** (Xu & Fu, 2025): We used the official repository with the same hyperparameters as in their benchmark. We observed, consistent with the authors' checkpoints in another benchmark, that the 20 sampled modes were nearly identical.

- **TranSPORTmer** (Capellera et al., 2024), **U2Diff** (Capellera et al., 2024): Official code and checkpoints were used for NBA. For NFL and Bundesliga, we trained with the same configuration. The authors provided the original codebase.

## D.2 COMPARISONS

### D.2.1 HUMAN-BASED METRICS

For each of the first 128 past observations in our test set, we formed a random pair from these selected samples. An example of the interface is shown in Fig. 7.

### D.2.2 INDIVIDUAL-BASED METRICS

This section provides a quantitative analysis using individual-based metrics ADE and FDE, reported as both minimum ($min$) and average ($avg$) values over 20 generated modes. Note that the average metrics are equivalent to SADE and SFDE as presented in Table 1 of the main paper.

Individual metrics comparison is depicted in Table 4. Our approach remains competitive in terms of $min$ metrics against non-IID baselines while, as stated in the main paper, excels in the $avg$ metrics. Crucially, our findings highlight a key divergence: the widely used $min$ ADE/FDE metrics do not correlate with the human evaluation from the previous section. For example, on the NBA dataset, MoFlow achieves the state-of-the-art in $min$ ADE / FDE, yet our human study showed that participants consistently preferred our approach. This suggests that SADE / SFDE metrics provide a more reliable indication of perceptual quality than ADE / FDE metrics Casas et al. (2020).

### D.2.3 COMPLETION GENERATION

We benchmark our approach on general trajectory completion using the datasets and experimental setup from Xu & Fu (2025), which includes the Basketball-U, Football-U, and Soccer-U datasets.

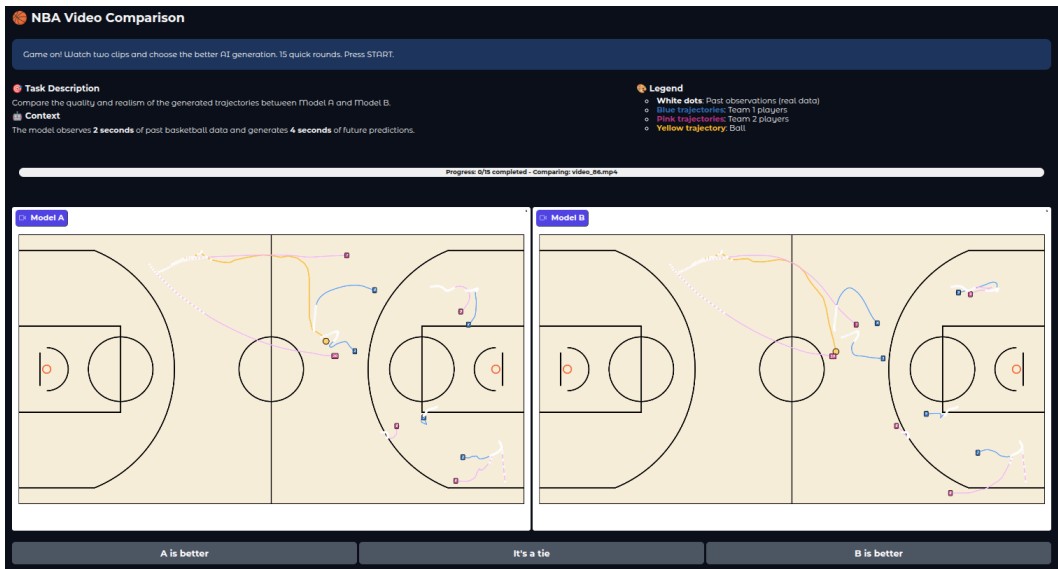

Figure 7: **Human Evaluation Interface.**

Table 4: **Future Generation.** We report ADE and FDE metrics ($min$ / $avg$) computed over 20 generated modes.

| Method | IID | NFL (yards) | | Bundesliga (meters) | | NBA (meters) | |
| --- | --- | --- | --- | --- | --- | --- | --- |
| | | ADE ↓ | FDE ↓ | ADE ↓ | FDE ↓ | ADE ↓ | FDE ↓ |
| GroupNet CVPR22 | ✓ | 1.70 / 5.33 | 3.19 / 12.18 | 1.89 / 5.76 | 3.23 / 11.63 | 0.94 / 2.84 | 1.22 / 5.15 |
| AutoBots ICLR22 | ✗ | 1.82 / 4.82 | 3.23 / 10.68 | 2.07 / 5.93 | 2.94 / 11.46 | 1.19 / 2.73 | 1.55 / 4.71 |
| LED$^{\text{IID}}$ CVPR23 | ✓ | 1.65 / 4.12 | 2.08 / 9.63 | 2.06 / 4.57 | 3.17 / 9.74 | 0.92 / 2.30 | 1.18 / 4.45 |
| LED CVPR23 | ✗ | - | - | - | - | 0.81 / 3.83 | 1.10 / 6.03 |
| MART ECCV24 | ✗ | 1.07 / 4.26 | 1.96 / 10.31 | **1.41** / 4.16 | **2.48** / 9.00 | 0.72 / 2.46 | 0.90 / 4.78 |
| MoFlow CVPR25 | ✗ | **1.03** / 4.02 | **1.87** / 9.98 | 1.47 / 4.21 | 2.74 / 9.24 | **0.71** / 2.42 | **0.86** / 4.64 |
| U2Diff CVPR25 | ✓ | 1.40 / 3.74 | 2.67 / 9.02 | 1.69 / 4.21 | 3.11 / 9.44 | 0.85 / 2.12 | 1.11 / 4.14 |
| JointDiff (Ours) | ✓ | 1.31 / **3.40** | 2.49 / **8.40** | 1.46 / **3.66** | 2.56 / **8.29** | 0.80 / **2.01** | 1.09 / **3.95** |

In this task, a pre-defined mask using different strategies is applied to a scene, and the model must complete the missing observations. We note that while our NFL dataset uses the same data splits as Football-U, it employs a different masking strategy. For comparison, we reuse the evaluation table from Xu & Fu (2025), also reported in Capellera et al. (2025), which includes a wide range of baselines. These baselines span statistical methods: **Mean**, **Median**, **Linear Fit**; vanilla models: **LSTM** (Hochreiter & Schmidhuber, 1997), **Transformer** (Vaswani et al., 2017)); and advanced methods: **MAT** (Zhan et al., 2019), **Naomi** (Liu et al., 2019), **INAM** (Qi et al., 2020), **SSSD** (Alcaraz & Strodthoff, 2022), **GC-VRNN** (Xu et al., 2023), **Sports-Traj** (Xu & Fu, 2024), and **U2Diff** (Capellera et al., 2025).

Results are reported in Table 5. We use the ADE metric as defined in Xu & Fu (2025), which we rename as **BADE** to reflect its dependence on batch size and distinguish it from the standard individual-level ADE or the scene-level SADE. The BADE metric is defined as:

$$\text{BADE} = \frac{\sum_{b=1}^{B} \sum_{n=1}^{N} \sum_{t=1}^{T} \left\| \hat{y}_{t,n}^{b} - y_{t,n}^{b} \right\|_2 \left(1 - m_{t,n}^{b}\right)}{\sum_{b=1}^{B} \sum_{n=1}^{N} \sum_{t=1}^{T} \left(1 - m_{t,n}^{b}\right)}, \tag{20}$$

where $y_{t,n}^{b}$ is the ground truth 2D spatial location of agent $n$ at timestep $t$ in scene $b$, $\hat{y}_{t,n}^{b}$ is its estimation, and $m_{t,n}^{b}$ is a value from the binary mask $\mathbf{M}$ where a value of 0 indicates a location to be predicted. In their setting, Xu & Fu (2025) use $B = 128$. We also report the standard SADE metric. The table presents the minimum ($min$) values across 20 generated modes. While the minimum mode for BADE is selected across the entire batch of scenes, the minimum mode for SADE is selected for each scene independently, making it independent of batch size and mode ordering.

Our approach notably outperforms previous baselines, showing a significant performance improvement in SADE of 13% in Basketball-U, 19% in Football-U, and 12% in Soccer-U. We observe that the provided checkpoints for the Sports-Traj model in this benchmark produced modes that were nearly-identical, with minimal differences between them.

To advance the field, in the main paper we advocate for the use of the widely-adopted NBA dataset as an alternative to Basketball-U. We have also curated a new Bundesliga dataset for soccer. This dataset offers significant advantages, as its sequences are substantially longer than those in Soccer-U, providing richer temporal context for modeling. In our curation, Bundesliga sequences last 6.4 seconds (40 frames at 6.25 fps), whereas the Soccer-U sequences, with 50 frames, last for two seconds or less.

Table 5: **Completion Generation.** We report the $min$ for BADE (ADE in (Xu & Fu, 2025)) and SADE over 20 generated modes.

| Method | Basketball-U (Feet) | | Football-U (Yards) | | Soccer-U (Pixels) | |
|---|---|---|---|---|---|---|
| | BADE ↓ | SADE ↓ | BADE ↓ | SADE ↓ | BADE ↓ | SADE ↓ |
| Mean | 14.58 | - | 14.18 | - | 417.68 | - |
| Median | 14.56 | - | 14.23 | - | 418.06 | - |
| Linear Fit | 13.54 | - | 12.66 | - | 398.34 | - |
| LSTM | 7.10 | - | 7.20 | - | 186.93 | - |
| Transformer | 6.71 | - | 6.84 | - | 170.94 | - |
| MAT | 6.68 | - | 6.36 | - | 170.46 | - |
| Naomi | 6.52 | - | 6.77 | - | 145.20 | - |
| INAM | 6.53 | - | 5.80 | - | 134.86 | - |
| SSSD | 6.18 | - | 5.08 | - | 118.71 | - |
| GC-VRNN | 5.81 | - | 4.95 | - | 105.87 | - |
| Sports-Traj | 4.77 | 4.29 | 3.55 | 4.03 | 94.59 | 100.48 |
| U2Diff | 4.65 | 3.13 | 2.42 | 2.35 | 53.93 | 51.14 |
| JointDiff (Ours) | **4.42** | **2.72** | **2.14** | **1.90** | **49.23** | **44.89** |

### D.2.4 FUTURE GENERATION

We further evaluate our method on future trajectory prediction using the basketball dataset, referred to as NBA$^{12/13}$, and protocol from Li et al. (2021). This setting enables comparison against temporal autoregressive baselines such as **NRI** (Kipf et al., 2018), **dNRI** (Graber & Schwing, 2020), and **GRIN** (Li et al., 2021); GAN-based models such as **Social-GAN** (Gupta et al., 2018); and a transformer-based approach, **FQA** (Kamra et al., 2020). Results are presented in Table 6, which reports metrics equivalent to our $min$ SADE and SFDE over 100 samples. JointDiff achieves strong performance relative to these forecasting models while operating as a more general trajectory completion framework, benefiting in particular from the non-autoregressive nature of diffusion along the temporal dimension.

Table 6: **Future Generation on NBA$^{12/13}$.** We report the $min$ for SADE and SFDE (ADE and FDE in (Li et al., 2021), respectively) over 100 generated modes.

| Method | NBA$^{12/13}$ (Feet) | |
|---|---|---|
| | SADE ↓ | SFDE ↓ |
| NRI | 2.10 | 4.56 |
| dNRI | 2.02 | 4.52 |
| FQA | 2.42 | 4.81 |
| Social-Gan | 1.88 | 3.64 |
| GRIN | 1.72 | 3.59 |
| JointDiff (Ours) | **1.36** | **2.52** |

### D.3 ABLATIONS

This section evaluates our method through three ablation studies: first, we examine the effect of varying the number of denoising steps $S^d$ in the discrete scheduler; second, we analyze the sensitivity of the $\lambda$ hyperparameter in our proposed joint loss (Eq. 8); and third, we investigate the contribution of additional components such as extra Social Transformers and importance sampling.

Table 7: **Denoising Step in Future Generation.** We report SADE and SFDE metrics ($min$ / $avg$) and Acc ($max$ / $avg$) over 20 generated modes.

| $S^d$ | $\zeta$ | NFL (yards) | | | Bundesliga (meters) | | | NBA (meters) | | |
|---|---|---|---|---|---|---|---|---|---|---|
| | | SADE ↓ | SFDE ↓ | Acc ↑ | SADE ↓ | SFDE ↓ | Acc ↑ | SADE ↓ | SFDE ↓ | Acc ↑ |
| 5 | 5 | 2.45 / 3.59 | 5.70 / 8.80 | .79 / .51 | 2.63 / 4.01 | 5.25 / 8.92 | .68 / .38 | 1.41 / 2.05 | 2.56 / 4.02 | .76 / .42 |
| | 10 | 2.43 / 3.46 | 5.61 / 8.37 | .75 / .47 | 2.74 / 4.23 | 5.41 / 9.19 | .64 / .35 | 1.40 / 2.00 | 2.55 / 3.90 | .74 / .38 |
| **10** | **5** | 2.36 / 3.40 | 5.53 / 8.40 | .78 / .54 | 2.47 / 3.66 | 5.02 / 8.29 | .68 / .39 | 1.39 / 2.01 | 2.53 / 3.95 | .75 / .45 |
| | 10 | 2.37 / 3.28 | 5.54 / 8.00 | .73 / .47 | 2.48 / 3.67 | 5.02 / 8.17 | .63 / .36 | 1.38 / 1.95 | 2.52 / 3.82 | .71 / .40 |
| 25 | 5 | 2.47 / 3.48 | 5.76 / 8.43 | .70 / .44 | 2.46 / 3.51 | 5.06 / 7.88 | .61 / .34 | 1.42 / 2.06 | 2.58 / 4.06 | .71 / .39 |
| | 10 | 2.66 / 3.70 | 6.05 / 8.61 | .53 / .30 | 2.45 / 3.43 | 5.00 / 7.63 | .53 / .28 | 1.40 / 1.96 | 2.55 / 3.86 | .63 / .32 |
| 50 | 5 | 2.57 / 3.60 | 5.95 / 8.57 | .55 / .31 | 2.53 / 3.73 | 5.13 / 8.29 | .54 / .28 | 1.43 / 2.09 | 2.60 / 4.14 | .66 / .35 |
| | 10 | 2.65 / 3.62 | 6.07 / 8.50 | .37 / .19 | 2.54 / 3.73 | 5.09 / 8.26 | .45 / .23 | 1.40 / 1.99 | 2.57 / 3.95 | .54 / .27 |

### D.3.1 DISCRETE DENOISING STEPS

We present an ablation study on the impact of the total discrete denoising steps ($S^d$) and the DDIM skipping parameter ($\zeta$). The total number of continuous denoising steps is fixed to $S = 50$ and we train our approach with different discrete denoising steps $S^d \in \{5, 10, 25, 50\}$. At inference we generate the samples with two different skipping intervals $\zeta \in \{5, 10\}$. This results in either 11 total steps ($\zeta = 5$) or six total steps ($\zeta = 10$). The results in Table 7 indicate that a configuration of $S^d = 10$ and $\zeta = 5$ is optimal for the reported SADE, SFDE, and Accuracy metrics.

### D.3.2 LAMBDA IN JOINT LOSS

We conducted an ablation study on the weighting hyperparameter, $\lambda$, in our joint loss function (Eq. 8). Our goal was to identify the largest value for $\lambda$ that does not negatively impact the quality of the continuous trajectory generation. Table 8 shows the results for various $\lambda$ configurations: $\lambda \in \{0, 0.001, 0.01, 0.1, 1\}$. Note that the case where $\lambda = 0$ is equivalent to our ablated model, Ours w/o joint. We found that $\lambda = 0.1$ represents the optimal trade-off, providing the most significant benefits from the joint modeling without overwhelming the primary trajectory generation task.

Table 8: **Lambda Sensitivity in Future Generation.** We report SADE and SFDE metrics ($min$ / $avg$) and Acc ($max$ / $avg$) over 20 generated modes.

| $\lambda$ | NFL (yards) | | | Bundesliga (meters) | | | NBA (meters) | | |
|---|---|---|---|---|---|---|---|---|---|
| | SADE ↓ | SFDE ↓ | Acc ↑ | SADE ↓ | SFDE ↓ | Acc ↑ | SADE ↓ | SFDE ↓ | Acc ↑ |
| 0 | 2.42 / 3.57 | 5.67 / 8.72 | .76 / .52 | 2.60 / 3.99 | 5.30 / 8.95 | .67 / .44 | 1.46 / 2.13 | 2.64 / 4.19 | .74 / .44 |
| 0.001 | 2.73 / 3.87 | 6.27 / 9.16 | .50 / .23 | 2.67 / 4.20 | 5.31 / 9.15 | .53 / .29 | 1.48 / 2.16 | 2.66 / 4.19 | .69 / .38 |
| 0.01 | 2.56 / 3.69 | 5.95 / 8.87 | .68 / .37 | 2.46 / 3.61 | 5.03 / 8.10 | .62 / .35 | 1.42 / 2.07 | 2.59 / 4.08 | .73 / .42 |
| **0.1** | 2.36 / 3.40 | 5.53 / 8.40 | .78 / .54 | 2.47 / 3.66 | 5.02 / 8.29 | .68 / .39 | 1.39 / 2.01 | 2.53 / 3.95 | .75 / .45 |
| 1 | 2.61 / 3.74 | 5.95 / 8.97 | .79 / .56 | 2.71 / 3.91 | 5.45 / 8.64 | .69 / .39 | 1.44 / 2.05 | 2.62 / 4.00 | .76 / .47 |

### D.3.3 GUIDANCE STRENGTH

As stated in the main paper, during inference we utilize a single forward pass corresponding to the conditioned output, which corresponds to using $w = 0$ in CFG (Ho & Salimans, 2022). With our DDIM sampling, this choice yields optimal performance, while extreme values degrade results. This setting also avoids the extra cost of dual forward passes. In Table 9 we report the same SADE / SFDE metrics as in Table 2 with $\mathcal{G}_{\text{text}}$ for NFL and Bundesliga datasets. CFG training enables a unified model supporting both controllable ($w = 0$) and non-controllable ($w = -1$) generation without notable degradation compared to our dedicated non-controllable model (Table 2, Ours w/o $\mathcal{G}$).

### D.3.4 T5 ENCODER

We compared the performance using T5-Small, T5-Base, and T5-Large encoders and observed only minor performance differences across the board. As shown in Table 10, T5-Base offers the best overall balance, achieving the lowest $min$ / $avg$ SADE and SFDE metrics on the NFL dataset, and competitive results on Bundesliga.

Table 9: **Guidance Weight** $w$**.** We report SADE and SFDE metrics ($min$ / $avg$) over 20 generated modes.

| $w$ | NFL (yards) | | Bundesliga (meters) | |
|---|---|---|---|---|
| | SADE ↓ | SFDE ↓ | SADE ↓ | SFDE ↓ |
| -1.0 | 2.44 / 3.47 | 5.75 / 8.59 | 2.51 / 3.70 | 5.10 / 8.33 |
| -0.5 | 2.21 / 3.11 | 5.09 / 7.58 | 2.12 / 2.84 | 4.19 / 6.06 |
| 0.0 | **2.19** / 3.09 | **5.04** / 7.52 | **2.08 / 2.72** | **4.09** / 5.68 |
| 0.5 | **2.19** / 3.10 | **5.04** / 7.56 | 2.12 / 2.73 | 4.18 / **5.66** |
| 1.0 | 2.21 / 3.15 | 5.10 / 7.68 | 2.21 / 2.81 | 4.40 / 5.84 |
| 2.0 | 2.31 / 3.31 | 5.34 / 8.07 | 2.49 / 3.07 | 5.05 / 6.41 |
| 3.0 | 2.46 / 3.53 | 5.69 / 8.58 | 2.80 / 3.38 | 5.71 / 7.04 |

T5-Large performs slightly worse than T5-Base, which is likely due to the stronger dimensionality compression required to map its $d = 1024$ dimensional embeddings down to our model's 256-dimensional hidden space. Overall, the encoder size has minimal impact on the model's performance, but the results suggest that excessive compression of a very large embedding space can lead to a slight degradation in trajectory prediction accuracy.

Table 10: **T5 Encoder Size.** We report SADE and SFDE metrics ($min$ / $avg$) for different T5 encoder sizes ($d$), measured over 20 generated modes.

| T5 Encoder ($d$) | NFL (yards) | | Bundesliga (meters) | |
|---|---|---|---|---|
| | SADE ↓ | SFDE ↓ | SADE ↓ | SFDE ↓ |
| Small (512) | 2.22 / 3.12 | 5.12 / 7.59 | 2.08 / 2.77 | **4.06** / 5.82 |
| Base (768) | **2.19 / 3.09** | **5.04 / 7.52** | **2.08 / 2.72** | 4.09 / **5.68** |
| Large (1024) | 2.24 / 3.25 | 5.14 / 7.88 | 2.10 / 2.75 | 4.11 / 5.74 |

### D.3.5 MULTINOMIAL VS ABSORBING

In the main paper, we compare the consistency of our multinomial discrete diffusion model with the absorbing state formulation in Table 3. To provide a more comprehensive assessment of generation quality, we extend this analysis using the same metrics as in Table 2 for controllable future generation. The results, presented in Table 11, compare our method when replacing the multinomial parameterization with an absorbingstate one (Ours w absorbing) against our original model. Across all tasks, the absorbing formulation exhibits consistently lower performance, which we attribute to reduced consistency between the generated trajectories and discrete events.

Table 11: **Controllable Generation with Absorbing.** This table compares our JointDiff model (using multinomial diffusion) against an ablation using absorbing state diffusion for discrete events (Ours w absorbing). As in Table2, we report the same metrics for both non-controllable (w/o $\mathcal{G}$) and controllable (w $\mathcal{G}_{\text{WPG}}$, w $\mathcal{G}_{\text{text}}$) future generation tasks.

| Method | NFL (yards) | | | Bundesliga (meters) | | | NBA (meters) | | |
|---|---|---|---|---|---|---|---|---|---|
| | SADE ↓ | SFDE ↓ | Acc ↑ | SADE ↓ | SFDE ↓ | Acc ↑ | SADE ↓ | SFDE ↓ | Acc ↑ |
| Ours w absorbing | | | | | | | | | |
| w/o $\mathcal{G}$ | 2.50 / 3.67 | 5.78 / 8.84 | .76 / .54 | 2.72 / 4.13 | 5.39 / 8.93 | .64 / .37 | 1.45 / 2.10 | 2.62 / 4.07 | .73 / .44 |
| w $\mathcal{G}_{\text{WPG}}$ | 2.57 / 3.84 | 5.72 / 8.92 | .83 / .65 | 2.36 / 3.44 | 4.43 / 7.04 | .75 / .49 | 1.28 / 1.89 | 2.25 / 3.66 | .87 / .67 |
| w $\mathcal{G}_{\text{text}}$ | 2.41 / 3.48 | 5.46 / 8.29 | .85 / .72 | 2.29 / 3.21 | 4.34 / 6.44 | .79 / .57 | - | - | - |
| Ours | | | | | | | | | |
| w/o $\mathcal{G}$ | 2.36 / 3.40 | 5.53 / 8.40 | .78 / .54 | 2.47 / 3.66 | 5.02 / 8.29 | .68 / .39 | 1.39 / 2.01 | 2.53 / 3.95 | .75 / .45 |
| w $\mathcal{G}_{\text{WPG}}$ | 2.29 / 3.26 | 5.29 / 7.94 | .84 / .65 | 2.13 / 2.85 | 4.22 / 6.16 | .77 / .52 | 1.24 / 1.81 | 2.20 / 3.53 | .87 / .67 |
| w $\mathcal{G}_{\text{text}}$ | 2.19 / 3.09 | 5.04 / 7.52 | .86 / .74 | 2.08 / 2.72 | 4.09 / 5.68 | .80 / .59 | - | - | - |

### D.3.6 ADDITIONAL ANALYSIS

We present the last ablation analysis which includes the relevance of two important components in our approach. The first one is the importance sampling from Nichol & Dhariwal (2021), and the second is the number of Social Transformers layers within residual denoising block. Table 12 shows the results of our approach without the importance sampling ("w/o IS") and with only one Social Transformer layer ("w 1 × ST"). Notice that the importance sampling is crucial when the size of

the dataset is small, like the NFL and the Bundesliga. Using two Social Transformers also improve the results across the three datasets.

Table 12: **Additional Ablation in Future Generation.** We report SADE and SFDE metrics ($min$ / $avg$) and Acc ($max$ / $avg$) over 20 generated modes.

| Method | NFL (yards) | | | Bundesliga (meters) | | | NBA (meters) | | |
|---|---|---|---|---|---|---|---|---|---|
| | SADE ↓ | SFDE ↓ | Acc ↑ | SADE ↓ | SFDE ↓ | Acc ↑ | SADE ↓ | SFDE ↓ | Acc ↑ |
| w/o IS | 2.74 / 3.78 | 6.28 / 9.09 | .75 / .50 | 2.78 / 4.03 | 5.51 / 8.87 | .65 / .36 | 1.44 / 2.05 | 2.61 / 4.00 | .74 / .44 |
| w 1 × ST | 2.46 / 3.54 | 5.71 / 8.59 | .78 / .52 | 2.54 / 3.69 | 5.18 / 8.32 | .66 / .38 | 1.44 / 2.07 | 2.63 / 4.05 | .74 / .43 |
| Ours | 2.36 / 3.40 | 5.53 / 8.40 | .78 / .54 | 2.47 / 3.66 | 5.02 / 8.29 | .68 / .39 | 1.39 / 2.01 | 2.53 / 3.95 | .75 / .45 |

## D.4    INTERPRETABILITY: ATTENTION ENTROPY

To understand how our joint modeling influences the learned correlations between agents, we analyzed the attention entropy of our Social Transformer layers. Solving the future generation task in the NBA dataset, we generated 20 modes from a batch of 128 using both our JointDiff (Ours) model and the ablated Ours w/o joint variant. We computed the entropy of the attention distribution for each agent at every layer and timestep. This entropy is defined as:

$$H(P) = -\sum_n P(x_n) \log_2 P(x_n), \tag{21}$$

where $P$ is the attention distribution, and $x_n$ is the $n$-th element in the set of agents (including the agent itself) over which the attention is computed. We then averaged this entropy across all attention heads, all four Social Transformer layers, all timesteps ($T$), all agents ($N$), and all 20 modes to obtain a single entropy value for each denoising step $s$. This per-step entropy provides a measure of the uniformity of the attention patterns, with higher entropy indicating more uniform (less focused) attention and lower entropy indicating more specialized (highly focused) attention on specific agents.

The difference in this averaged entropy between the two models is depicted in Fig. 8. Our analysis reveals that JointDiff maintains a consistently lower attention entropy. The gap is most pronounced during the initial denoising steps, suggesting that agents in our model attend in a more focused manner from the beginning. This supports our hypothesis that providing the model with the possessor event allows it to immediately identify and prioritize the most salient interactions in the scene.

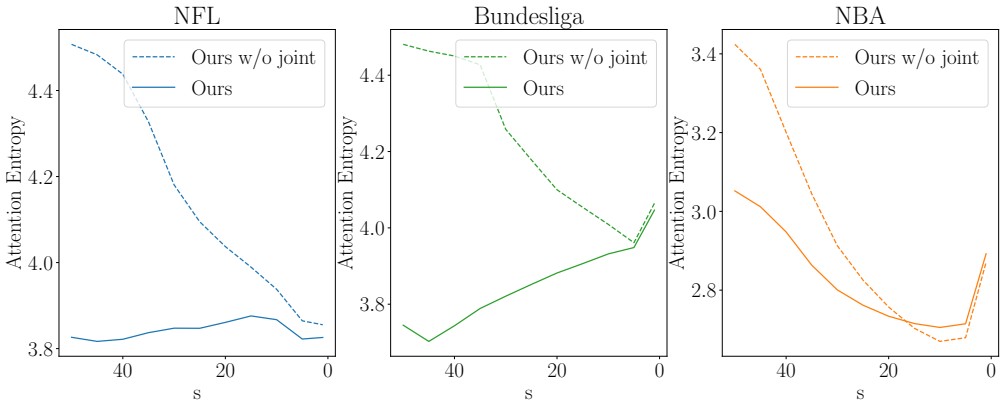

Figure 8: **Entropy vs Denoising Step.** We report the entropy of the inter-agent attention across the three datasets.

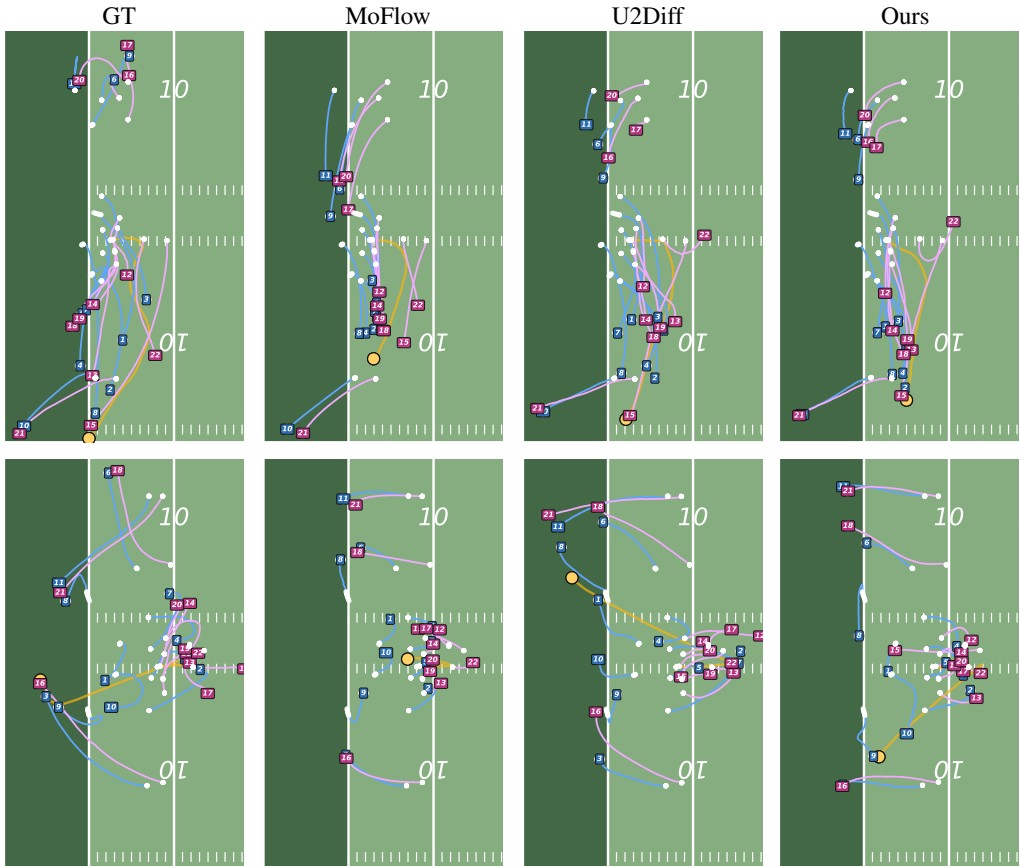

Figure 9: **Future Generation on NFL.** Comparison of Ours vs. MoFlow and U2Diff baselines on generating future 40 timesteps conditioned on 10 past observed timesteps. Legend: ● Ball, ■ Home team, ■ Away team, ○ Past observations.

## D.5 QUALITATIVE RESULTS

We provide further qualitative results in completion generation and controllable generation, while also showing failure cases. Please refer to the video supplementary to see animated results.

### D.5.1 COMPLETION GENERATION

This section provides qualitative comparisons of JointDiff (Ours) against the state-of-the-art generative models MoFlow and U2Diff. We depict the mode with the best SADE metric over 20 generated modes in Fig.9 for the NFL, in Fig.10 for the Bundesliga, and in Fig.11 for the NBA.

### D.5.2 CONTROLLABLE GENERATION

This section presents qualitative results for the controllable future generation task. For weak-possessor-guidance (Fig. 12), we compare a single generated mode from our full model (Ours) against the variant without joint training (Ours w/o joint). For text-guidance (Fig. 13), we generate 20 modes for our method and qualitatively select the sample most aligned with the text description.

### D.5.3 FAILURE CASES

A key limitation observed in our results is the occasional inconsistency between the generated trajectories and the text-guidance. This can be traced to the constrained size of our training datasets, $\approx$ 10k pairs for NFL and $\approx$ 2k (augmented to 4k) for Bundesliga. The limited data variety hinders

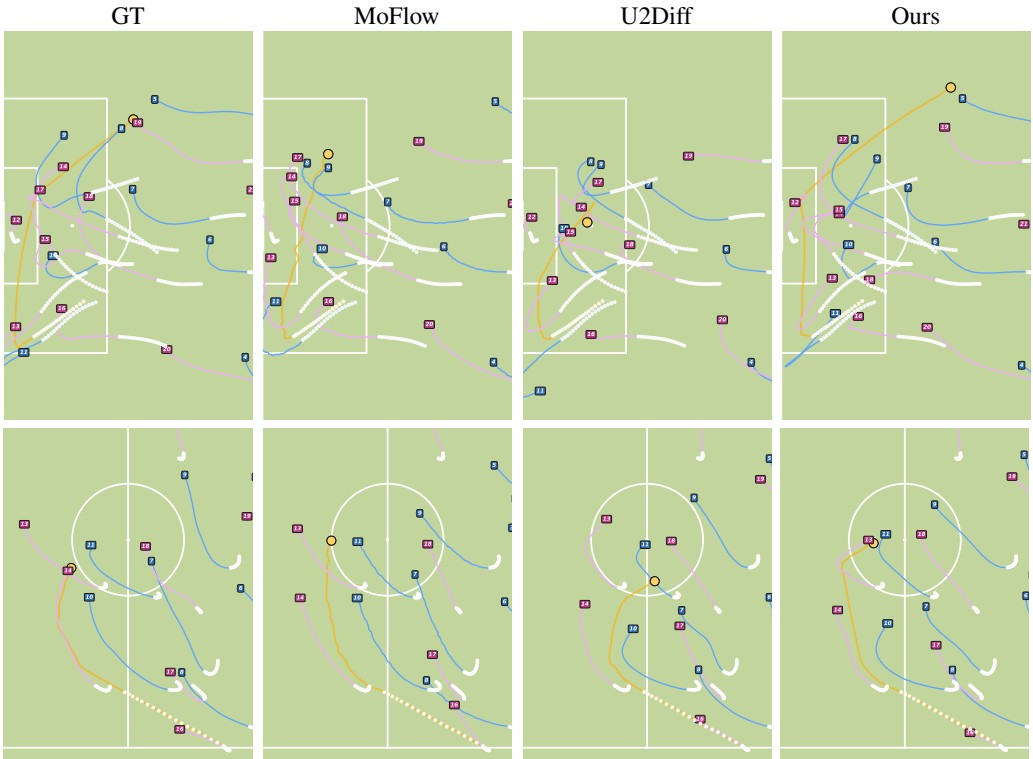

Figure 10: **Future Generation on BundesLiga.** Comparison of Ours vs. MoFlow and U2Diff baselines on generating future 30 timesteps conditioned on 10 past observed timesteps. Legend: ● Ball, ■ Home team, ■ Away team, ○ Past observations.

the model's ability to robustly encode the wide range of possibilities described in natural language. Refer to Fig.14 to see some failure examples from the same scenarios depicted in Fig. 13.

## E    LIMITATIONS AND FUTURE WORK

Our model's architecture requires events to share the same spatio-temporal structure as the trajectory data, i.e., to allow for simple concatenation at the input. This limits its application to event streams that are naturally structured this way and cannot directly handle unstructured data, such as sparse temporal point processes. A key direction for future work is to develop methods to integrate these more complex event types.

A second limitation stems from the NFL dataset. The public event data does not identify the player responsible for each action. Consequently, we must rely on a combination of heuristics and tracking data to assign an actor, a process that may lead to sub-optimal outcomes.

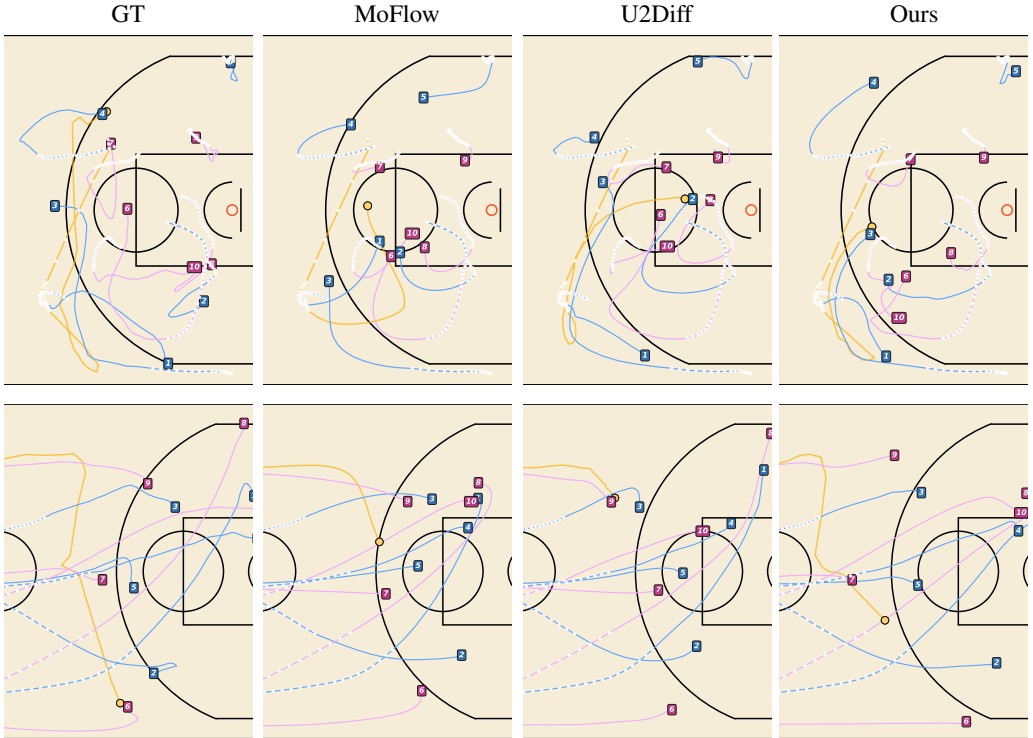

Figure 11: **Future Generation on NBA.** Comparison of Ours vs. MoFlow and U2Diff baselines on generating future 20 timesteps conditioned on 10 past observed timesteps. Legend: 🟡 Ball, 🟦 Home team, 🟪 Away team, ◯ Past observations.

# F    STATEMENTS

## F.1    LLM USAGE

LLMs were used in two ways: (1) to improve the grammar and readability of the manuscript, and (2) to post-process the generated text dataset by correcting grammar and ensuring consistency (as described in the paper). All aspects of the research design, modeling, experimentation, and analysis were carried out independently of any LLM assistance.

## F.2    ETHICS

This research uses trajectory data representing human agents. All datasets employed are either publicly available or synthetically generated, and contain no personally identifiable information. The trajectories and textual descriptions are anonymized and represent abstract positions rather than identifiable individuals (as described in the paper). The intended applications of this work include sports analytics and multi-agent simulation, which we believe pose minimal ethical risk.

## F.3    DATASET

We commit to releasing the dataset and the code necessary to reproduce it upon acceptance of this paper for publication.

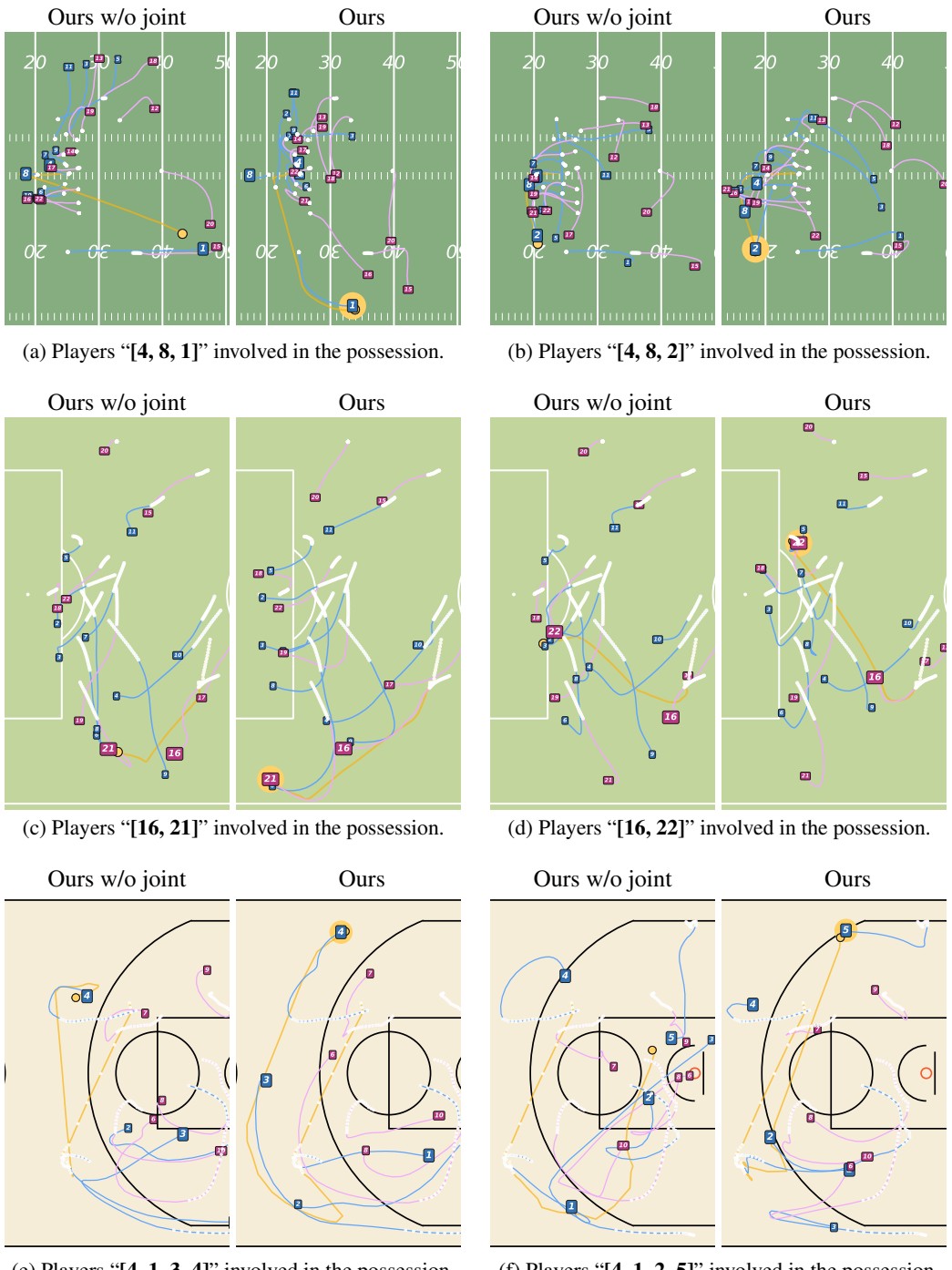

Figure 12: **Controllable Generation with WPG.** Comparison of Ours vs. Ours w/o joint on the weak-possesor-guidance task giving the same past observations with different possessor sequences $\mathcal{G}_{\text{WPG}}$. Legend: ● Ball, ■ Home team, ■ Away team, ○ Past observations.

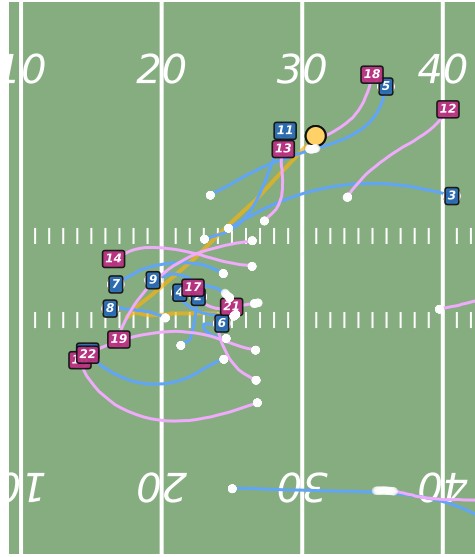 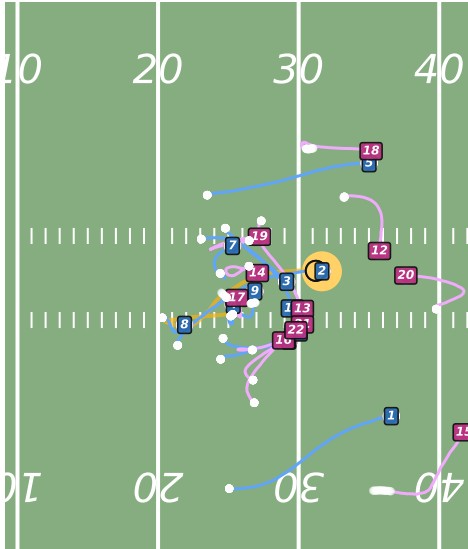

(a) "Home Team has possession in SHOTGUN formation. Player 4 snaps the ball to Player 8 at yard L 25. Player 8 possesses the ball and throws a forward pass to Player 5. The ball travels from yard L 15 to L 30."

(b) "Home Team has possession in SHOTGUN formation. Player 4 snaps the ball to Player 8 at yard L 25. Player 8 makes a hand-off pass to Player 2 and he runs with the ball."

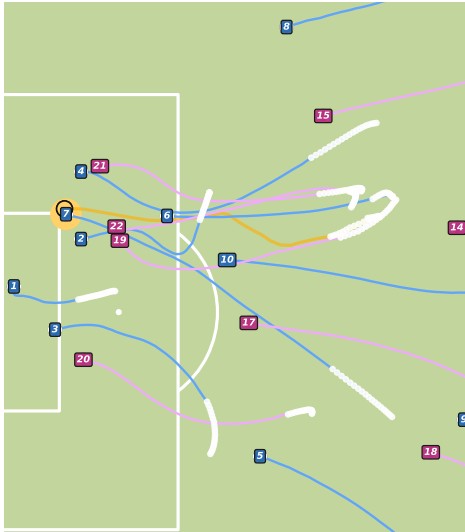 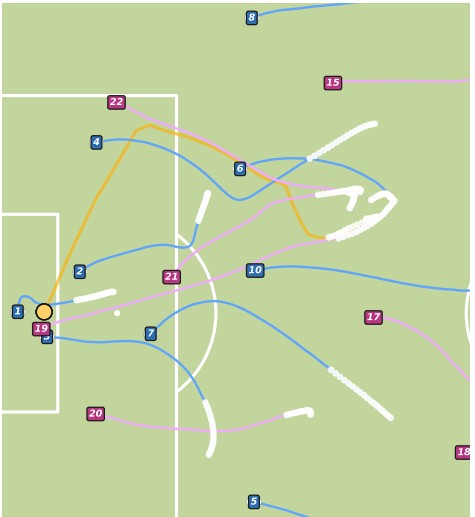

(c) "Away Team has the possession. The ball starts at left-center. Player 19 possesses the ball at left-center and passes to Player 22. The ball moves from left-center to box, then to up-corner. Player 22 possesses the ball at up-corner and attempts a pass, which is intercepted by Home Player 7. Home Team gains possession and Player 7 attempts a clearance, as the ball moves to box."

(d) "Away Team has the possession. The ball starts at left-center. Player 19 possesses the ball at left-center and passes to Player 22. The ball moves from left-center to box, then to up-corner. Player 22 possesses the ball at up-corner and attempts a pass to Player 19 inside the box."

Figure 13: **Controllable Generation with Text.** Examples on text-guidance task giving the same past observations with different prompts $\mathcal{G}_{\text{text}}$. Legend: ● Ball, ■ Home team, ■ Away team, ○ Past observations.

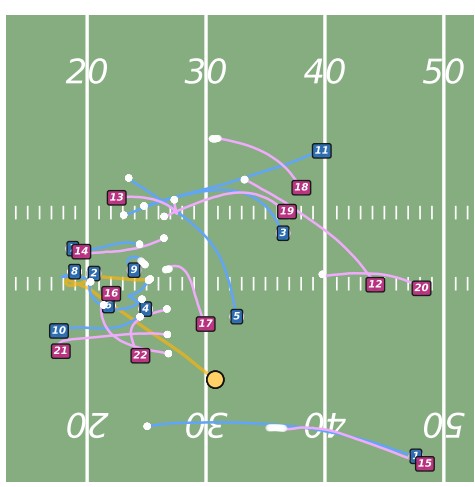 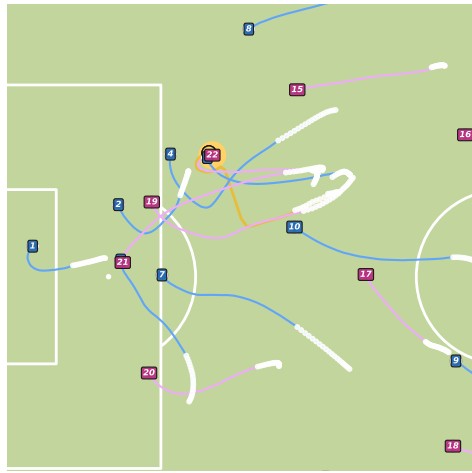

(a) "Home Team has possession in SHOTGUN formation. Player 4 snaps the ball to Player 8 at yard L 25. Player 8 possesses the ball and throws a forward pass to Player 5. The ball travels from yard L 15 to L 30."

(b) "Away Team has the possession. The ball starts at left-center. Player 19 possesses the ball at left-center and passes to Player 22. The ball moves from left-center to box, then to up-corner. Player 22 possesses the ball at up-corner and attempts a pass to Player 19 inside the box."

Figure 14: **Failure Cases on Controllable Generation with Text.** Legend: 🟡 Ball, 🟦 Home team, 🟪 Away team, ⬭ Past observations.

