# OpenReview forum: "JointDiff: Bridging Continuous and Discrete in Multi-Agent Trajectory Generation"
_ICLR.cc/2026/Conference — ICLR 2026 Poster_

### Official Review · Reviewer_nw5t · 2025-10-26

**Soundness:** 3
**Presentation:** 3
**Contribution:** 3
**Rating:** 6
**Confidence:** 3

**Summary:**

The paper describes generating multi-agent trajectories in a sports scene, ensuring that the spatio-temporal trajectories of players and the ball are plausible. Motivated by the success of the recent diffusion-based generative framework, the paper proposes combining a discrete representation (ball possession) with a continuous representation (spatial coordinates of individual trajectories). Additionally, the framework can provide additional guidance of user intent, composed of text guidance or weak processor guidance. The results are presented with soccer and football datasets.

**Strengths:**

- The work proposes an important yet challenging problem, which generates controllable multi-agent trajectories. Theoretically, the framework combines the continuous and discrete representations in a consistent way and can be extended to many other large-scale control systems.

**Weaknesses:**

- Figures are hard to interpret. All the texts are too small, and the terminology is not clearly defined. Things to be in a clear layout with large enough text whose sizes are comparable to the main paper.

- The exposition in Section 4.3 is weaker compared to other parts of the paper. It is hard to see how different components are put together, and the limited clarity of the figure does not help in understanding the structure. I suggest adding a simplified version of Figure 4 in the main paper.

**Questions:**

Minor comment
- temporal-evolving -> temporally evolving (line 051)
- Why do you call the works non-generative in line 092?

---

> ### Author Response · Authors · 2025-11-19
>
> We thank the reviewer for their positive assessment of our work's theoretical consistency and potential for broad application. We are grateful for their constructive feedback on improving the paper's presentation.
>
> ### On Figure Clarity (Weakness 1)
> We appreciate the reviewer's feedback on figure clarity. We will address this in the final version by increasing font size in figures to improve the text readability. We also revise the terminology to be clearly defined. These revisions will substantially improve the readability and clarity of our figures.
>
>
> ### On Section 4.3 Exposition (Weakness 2)
> Thanks for your suggestion regarding the section 4.3. Our idea was to illustrate a simplified version of the Figure 4 in the Figure 2-left, and on the right illustrate our proposed CrossGuid, inside the Social-Temporal Block. We try not to overload the main part with other operations rather than the Social-Temporal Block and CrossGuid. On the other hand, in Figure 4 we show all operations in the architecture and defining them in Appendix B. According to the reviewer's suggestion, we will further enhance the mapping between the Figure 2-left and Figure 4.
>
>
> ### Minor comments
> * Line 051 (Typo): We thank the reviewer for spotting this. We will correct "temporal-evolving" to "temporally evolving" in the final manuscript.
> * Line 092 ("non-generative"): In line 092 we used "non-generative" to architectures that produce deterministic outputs rather than sampling from prior distributions. We can clarify this by rephrasing to:
> "More recently, non-sampling based methods, tipically based on transformer architectures (Vaswani et al.,2017), sometimes enhanced with visual data (...), have significantly improved the prediciton of ...".
>
> We thank the reviewer again for the supportive feedback. Incorporating the suggested clarifications will help us strengthen the paper.

---

### Official Review · Reviewer_Q4Be · 2025-10-27

**Soundness:** 3
**Presentation:** 4
**Contribution:** 3
**Rating:** 6
**Confidence:** 4

**Summary:**

The paper propose Jointdiff, a diffusion-based framework tailored for trajectory generation in ball games. Jointdiff includes social-temporal block that models continuous multi-agent trajectory and discrete events in a single denoising process. Controllable generation is enabled via Crossguid, which injects external guidance signal represented by . The guidance could be weak-possessor-guidance or text descriptions. Jointdiff is evaluated with three types of ball games-- NBA, NFL and Bundesliga. Results shows Jointdiff achieves SOTA with respect to future prediction and imputation.

**Strengths:**

1. Principled diffusion-based joint modeling of continuous and discrete modalities.
2. Controllable scene generation enable interesting applications, for example, it could simulates the outcome of different tactics, to make better decisions.
3. Evaluation across three types of sports
4. Supplementary materials includes many details about text guidance dataset generation, which could be useful for constructing text dataset for other use cases.
5. Model architecture graph is very clear.

**Weaknesses:**

1. The main paper presents a broadly applicable diffusion architecture but offers limited methodological novelty; the contribution reads more application-driven than algorithmic. Key elements that could strengthen the perceived innovation—e.g., human evaluation design, text-dataset construction, and other data curation details—are largely relegated to the supplementary material. As a result, the main paper feels incremental and under-substantiated. The authors should write more these components into the main text, clearly describe what is genuinely new versus adapted, and include ablations that isolate architectural contributions from application engineering.

2. The aligned training method of text and trajectory is not clear.

3. The related-work discussion is thin and misses several foundational lines (e.g., NRI, dNRI, GRIN)[1,2] and key domain papers (e.g., TacticAI)[3], which weakens the paper’s positioning and novelty claim; at minimum, the authors should (i) contextualize how JointDiff differs conceptually, and (ii) add targeted comparisons/ablations or a rationale for incompatibilities.

[1Kipf, T., Fetaya, E., Wang, K. C., Welling, M., & Zemel, R. (2018, July). Neural relational inference for interacting systems. In International conference on machine learning (pp. 2688-2697). Pmlr.

[2]Li, L., Yao, J., Wenliang, L., He, T., Xiao, T., Yan, J., ... & Zhang, Z. (2021). Grin: Generative relation and intention network for multi-agent trajectory prediction. Advances in Neural Information Processing Systems, 34, 27107-27118.

[3]Wang, Z., Veličković, P., Hennes, D., Tomašev, N., Prince, L., Kaisers, M., ... & Tuyls, K. (2024). TacticAI: an AI assistant for football tactics. Nature communications, 15(1), 1906.

**Questions:**

1. How exactly is text guidance trained? Please specify the text encoder (e.g., frozen vs. finetuned), how its embeddings are injected (e.g., CrossGuid placement, keys/values/queries), and whether you use classifier-free guidance during training (drop-condition rate, loss formulation).
2. Do you align text to trajectories/events with any auxiliary objectives (e.g., contrastive losses) or just condition the denoiser? Any ablations on encoder choice and guidance strength?
3. Please report the full compute profile: GPU model/count, batch size.
4. How controllable is the generation process? How is the sampling efficiency?
5. How to deal with uncertainties in future generation? e.g., their might be multiple possible future scenarios given the same historical trajectory?
6. Can the model be used to review the middle of a match? For example, a tactical switch or a player substitution?
7. In your opinion, is there a foundation model that can model the trajectories of all ball games?

---

> ### Author Response · Authors · 2025-11-19
> **Part (1/2)**
>
> We thank the reviewer for their positive assessment and exceptionally constructive feedback. We agree that the paper's contribution and positioning can be significantly strengthened by incorporating more details from the supplement and expanding our related work, and we plan to do so in the final version.
>
> ### Positioning and Contribution (W1)
> We appreciate the reviewer’s perspective on our contribution. Our goal is to introduce a novel algorithmic framework, motivated and validated by the challenging setting of multi-agent sports. As detailed in our response to reviewer 5jzA, we outline the core differences from prior work.
>
> We also agree that the human evaluation and the construction of our augmented text descriptions are substantial contributions on their own. Following the reviewer’s suggestion, we will use the extra page in the final version to move these elements from the appendix into the main paper. This will better highlight the full scope of our contribution, which combines the joint diffusion framework with new controllable tasks and a corresponding dataset for this benchmark.
>
> ### Text settings (W2)
> We provide detailed clarifications in the Questions section below.
>
>
> ### Related Work and New Comparisons (W3)
> We thank the reviewer for providing these valuable citations. We will add a full discussion of this Related Work to the final manuscript.
>
> * NRI[1], dNRI [4], and GRIN [2]: These approaches are relevant within our domain but methodologically different: JointDiff is a non-autoregressive, diffusion-based generative model. In contrast, NRI[1]/dNRI[4] use variational autoencoders, and GRIN[3] uses a graph-recurrent architecture, with all three typically being autoregressive over time. Our diffusion approach generates full, long-horizon trajectories non-autoregressively. For a fair comparison, we evaluated JointDiff on the NBA 2012–2013 dataset using the exact protocol and SADE/SFDE metrics from GRIN[2] (they report the minimum over 100 samples, while ADE/FDE is equivalent to our SADE/SFDE). JointDiff performs strongly against these forecasting models while remaining a more general completion framework.
>
> | Model        | SADE  | SFDE  |
> |--------------|------|------|
> | NRI [1]      | 2.10 | 4.56 |
> | dNRI [4]     | 2.02 | 4.52 |
> | FQA [5]      | 2.42 | 4.81 |
> | Social-GAN [6]  | 1.88 | 3.64 |
> | GRIN [2]     | 1.72 | 3.59 |
> | JointDiff (Ours) | **1.36** | **2.52** |
>
> * TacticAI[3]: While valuable paper for static set-piece optimization, it addresses a different problem than our dynamic, open-play sequence generation framework.
>
> We plan to provide the additional experimental analysis suggested by the reviewer in the final version.
>
> [1] Neural relational inference for interacting systems. (ICML 2018) \
> [2] Grin: Generative relation and intention network for multi-agent trajectory prediction. (NeurIPS 2021) \
> [3] TacticAI: an AI assistant for football tactics. (Nature communications 2024) \
> [4] Dynamic neural relational inference. (CVPR 2020) \
> [5] Multi-agent trajectory prediction with fuzzy query attention.(NeurIPS 2020) \
> [6] Social gan: Socially acceptable trajectories with generative adversarial networks.(CVPR 2018)
>
> ### Questions
> **Q1: Text Guidance Training**
> We employ classifier-free guidance (CFG) during training, randomly dropping the condition $\mathcal{G}$ with a probability of 25% (l.259-260). During inference we utilize only a single forward pass corresponding to the conditioned output (l.261-262), which corresponds to using $w=0$ in the original CFG paper. We show to be effective in our domain, as shown in Q2, while avoiding duplicating the computational cost of dual forward passes.
>
> The CrossGuid module (Section 4.3.1) handles the conditioning. For text guidance, the text signal $\mathcal{G}_\text{text}$ is encoded using the frozen, pre-trained T5-Base encoder, producing the conditioning tensor $G \in \mathbb{R}^{L \times 768}$ (l.334-335). We will explicitly clarify in the main text that condition dropping ($\mathcal{G}=\emptyset$) is implemented by setting $G$ to a zero tensor. The subsequent text (lines 337-346) details the cross-attention mechanism, specifying how the Key-Value pairs and Query are constructed for text-guidance.

---

> > ### Author Response · Authors · 2025-11-19
> > **Part (2/2)**
> >
> > **Q2: Alignment and Ablations**
> > We do not use auxiliary alignment objectives; the denoiser is conditioned directly on the guidance signal through CrossGuid. Per the reviewer's request, we conducted two new ablations on the text-guidance task:
> >
> > 1. Guidance strength ($w$): Using $w=0$ with our DDIM sampling (as noted in Q1) yields optimal or near-optimal performance, while extreme values degrade results. This setting also avoids the extra cost of dual forward passes. We report the same SADE/SFDE metrics as Table 2 with $\mathcal{G}_\text{text}$:
> >
> > |              | NFL  |  (yards)     | Bundesliga  | (meters) |
> > |--------------|------|------|------|------|
> > |              | SADE  | SFDE  | SADE | SFDE |
> > | $w$ = -1.0     |  2.44 / 3.47  |  5.75 / 8.59   |  2.51 / 3.70  |  5.10 / 8.33   |
> > | $w$ = -0.5     |  2.21 / 3.11  |  5.09 / 7.58   |  2.12 / 2.84  |  4.19 / 6.06   |
> > | $w$ = 0.0      |  **2.19 / 3.09**  |  **5.04 / 7.52**   |  **2.08 / 2.72**  |  **4.09** / 5.68   |
> > | $w$ = 0.5      |  **2.19** / 3.10  |  **5.04** / 7.56   |  2.12 / 2.73  |  4.18 / **5.66**   |
> > | $w$ = 1.0      |  2.21 / 3.15  |  5.10 / 7.68   |  2.21 / 2.81  |  4.40 / 5.84   |
> > | $w$ = 2.0      |  2.31 / 3.31  |  5.34 / 8.07   |  2.49 / 3.07  |  5.05 / 6.41   |
> > | $w$ = 3.0      |  2.46 / 3.53  |  5.69 / 8.58   |  2.80 / 3.38  |  5.71 / 7.04   |
> >
> > CFG training enables a unified model supporting both controllable ($w=0$) and non-controllable ($w=-1$) generation without notable degradation compared to our dedicated non-controllable model (Table 2, Ours w/o $\mathcal{G}$).
> >
> > 2. T5 Encoder: We compared T5-Small, Base, and Large and found only minor performance differences. T5-Base offers the best balance. T5-Large is slightly worse, likely due to compressing its 1024-dimensional embeddings to our 256-dimensional hidden space. Below is the comparison (same setting as Table 2 with $\mathcal{G}_\text{text}$).
> >
> > |              | NFL  |  (yards)     | Bundesliga  | (meters) |
> > |--------------|------|------|------|------|
> > |              | SADE  | SFDE  | SADE | SFDE |
> > | Small (512)  |  2.22 / 3.12  |  5.12 / 7.59   |  2.08 / 2.77  |  **4.06** / 5.82   |
> > | Base (768)   |  **2.19 / 3.09**  |  **5.04 / 7.52**   |  **2.08 / 2.72**  |  4.09 / **5.68**   |
> > | Large (1024) |  2.24 / 3.25  |  5.14 / 7.88   |  2.10 / 2.75  |  4.11 / 5.74   |
> >
> > Overall, encoder size has minimal impact, with T5-Large slightly degraded due to the stronger dimensionality compression.
> >
> >
> > **Q3: Implementation Details**
> > As noted, batch size is 16 in all experiments (l.413–414). All models were trained on a single RTX A6000. Per-epoch times are ~1.00 / 0.50 / 2.20 minutes for NFL / Bundesliga / NBA.
> >
> >
> > **Q4: Controllability and Efficiency**
> > * Controllability: Both qualitative and quantitative evidence show strong controllability. The supplementary video (1:20–1:50) shows realistic rollouts adhering to WPG and text guidance. Table 2 confirms that conditioning on ground-truth guidance improves all metrics compared to non-controlled (w/o $\mathcal{G}$) generation.
> > * Efficiency: We use 11 DDIM steps. With $w=0$, controllable generation uses a single forward pass per step. CrossGuid adds minimal overhead; sampling is ~1.3× slower than the non-controlled setting.
> >
> > **Q5: Uncertainty Handling**
> > This is a key strength of our generative model. Our model inherently captures the one-to-many nature of future trajectories. We demonstrate this in the supplementary video (1:05-1:20), where we show multiple, diverse, and realistic futures all generated from the exact same past trajectory observation.
> >
> > **Q6: Tactical Analysis**
> > This is an insightful question. Our model can analyze tactical switches within its ~10-second horizon and supports rich “what-if’’ scenarios by conditioning not only on trajectories but also on strategic signals such as formations, possession sequences (via WPG), and text-based descriptions of gameplay. Analysts can adjust these inputs (altering formations, changing possession sequences, or providing alternative text prompts) to explore different strategic outcomes, identify potential scoring opportunities, or compare defensive responses. The model is player-agnostic and does not encode individual physical or technical attributes. Consequently, substituting players within the same role does not change the generated outcome, which depends solely on spatiotemporal and semantic context rather than player identity.
> >
> > **Q7: Foundation Model Potential**
> > We believe this is a promising direction to build a foundation model. Our framework's success across three different sports (with different rules, team sizes, and dynamics) suggests the core architecture is highly generalizable. A foundational model could likely be built, perhaps with an added "sport-type" conditioning signal, to learn a unified representation of multi-agent dynamics.

---

> ### Comment · Reviewer_Q4Be · 2025-11-20
> **Response of authors' comments**
>
> The authors have addressed most of my main concerns. The clarifications and additional evidence strengthen the contribution and resolve my earlier doubts. I am therefore increasing my overall score from 6 to 8.

---

> > ### Author Response · Authors · 2025-11-28
> >
> > We sincerely thank the reviewer for their thorough and constructive evaluation. We greatly appreciate the time invested in reassessing our work and for the positive update to the overall score.
> >
> > As suggested, we have incorporated the discussed references and expanded analyses into the revised manuscript, which we believe further strengthens the clarity and impact of our contributions.
> >
> > Thank you again for your helpful feedback and support.

---

### Official Review · Reviewer_5jzA · 2025-11-01

**Soundness:** 2
**Presentation:** 3
**Contribution:** 2
**Rating:** 4
**Confidence:** 3

**Summary:**

The paper proposes JointDiff, a diffusion-based generative model for multi-agent trajectories in sports. This method jointly models the continuous trajectories and discrete events in multi-agent systems, enhancing the generation quality. The authors also introduce two new controllable generation tasks: weak-possessor-guidance (WPG), which conditions on an ordered list of possessors, and "text-guidance," which conditions on natural language descriptions. To facilitate this, they propose an architectural module, CrossGuid, a cross-attention mechanism that injects the guidance signal into the diffusion model's denoising network. The authors validate their model on multiple datasets and release a new sports benchmark with enhanced text annotations. JointDiff demonstrates state-of-the-art results on completion/imputation tasks and demonstrating effective control in their new tasks. A human study is also provided, which shows a perceptual preference for the joint model's generations.

**Strengths:**

1. JointDiff is the first work to introduce joint continuos&discrete diffusion n modeling to time-evolving multi-agent systems. The authors also proposed new guidance methods for controllable generation and corresponding architecture (CrossGuid). In addition, a new sport benchmark is introduced.

2. The evaluation on sport datasets is comprehensive. Three public sports datasets, NBA, NFL, and BundesLiga are covered, along with two tasks, completion and controllable generation. Human study is also provided.

**Weaknesses:**

1. A major concern is about the novelty of the proposed method. As stated in the related work, a couple of prior work in other domains have introduced joint diffusion modeling of continuous and discrete data. Can the authors further illustrate how this work distinguish with them in addition to different task settings?

2. The CrossGuid module seems like a standard cross-attention block engineered for the sport multi-agent tasks.

**Questions:**

1. What is the main contribution beyond the two prediction heads design and the cross-attention gating? Why is this different from prior joint continuous–discrete diffusion work?

2. How does the model ensure consistency during generation? Since it learns to predict $Y_0$ and $E_0$ from two separate heads, is it possible for it to generate a final $(Y_0, E_0)$ pair that is inconsistent? Does this happen in practice?

---

> ### Author Response · Authors · 2025-11-19
> **Part (1/2)**
>
> We thank the reviewer for their thoughtful feedback and for raising important questions about novelty and consistency. We are confident we can resolve these concerns.
>
> ###  Core contribution
> The main contribution of JointDiff is the introduction and validation of a novel unified framework for dynamic multi-agent generation. Our novelty extends beyond the task setting to fundamental technical choices that are uniquely suited to the challenges of dynamic, interactive systems:
> * Multi-Agent, Temporal Domain: Unlike prior works on static data (layouts, images) or single sequences, we tackle the complex, two-dimensional problem of modeling the joint evolution of N agents over time.
> * Technical co-dependency: We pioneer the combination of multinomial discrete diffusion (enabling fine-grained event refinement [2, 3]) with a noise-prediction objective for continuous trajectories (producing high-fidelity paths [1]). This enables mutual, bidirectional conditioning: the denoising of trajectories is informed by the evolving event state, and in reverse.
> * Simultaneous Generation & Semantic Control: This specific combination, jointly with CrossGuid, enables true simultaneous generation and novel semantic control (WPG, Text), which we rigorously ablate.
>
> The table below provides a precise, at-a-glance comparison of how our technical formulation differs from the recent prior art:
>
> | Work | Data Domain | Discrete Diffusion | Continuous Diffusion | Denoising Strategy | Controllability |
> | :--- | :--- | :--- | :--- | :--- | :--- |
> | Levi et al. 2023 | Static (Layouts) | Absorbing-state | Predicts Data | Simultaneous | Layout Constraints (No Text) |
> | Zeng et al. 2024 | Dynamic, Single-Sequence | Multinomial | Variational Bound | Sequential (D→C) | No |
> | Li et al. 2025 | Static (Image+Text) | Absorbing-state | Predicts Noise | Separate Tasks (T2I/I2T) | Yes |
> | Chereddy et al. 2025 | Static (CAD Sketches) | Gaussian-Softmax | Predicts Data | Simultaneous | No |
> | **JointDiff (Ours)** | **Dynamic, Multi-Agent** | **Multinomial** | **Predicts Noise** | **Simultaneous** | **Yes (Novel: WPG & Text)** |
>
> Below we propose the extended version of this table and some justifications:
>
> While Levi et al. (2023) pioneered joint diffusion for static layouts, while JointDiff advances beyond this by tackling dynamic multi-agent systems and employing a more refined technical approach. We replace their absorbing-state discrete diffusion [3], which fixes tokens once unmasked, preventing further refinement [4], with multinomial diffusion that enables continuous refinement throughout all denoising steps. Furthermore, we predict noise for trajectories rather than clean data, yielding smoother temporal sequences. Finally, they use the standard DDPM sampler for continuous variables at inference time, while JointDiff adopts DDIM formulation at inference time for better sampling efficiency.
>
> Compared to Zeng et al. (2024), who model a single event sequence for future prediction, JointDiff tackles the more general task of completing (e.g. forecasting or imputing) the full scene by generating multi-agent trajectories together with synchronous events. While they use sequential denoising (discrete then continuous), we perform true simultaneous denoising with bidirectional conditioning. Our framework also enables novel controllability through weak-possession and text guidance, which their work does not explore.
>
> Unlike Li et al. (2025) who focus on static image-text pairs with separate generation tasks, JointDiff performs true simultaneous generation of both modalities in dynamic systems. While they require multi-stage training for different tasks, our approach trains end-to-end for unified generation. We also differ beyond their absorbing-state diffusion with our multinomial formulation.
>
> Finally, while Chereddy & Fermiani (2025) apply joint diffusion to static CAD sketches, JointDiff addresses dynamic multi-agent temporal sequences. We differ technically in using noise prediction for continuous variables and multinomial diffusion for discrete events, unlike their clean-data prediction and Gaussian-softmax approach. Our work also pioneers controllable generation in this domain.
>
> This specific configuration is not an incremental change; it is a purposeful architecture that enables the demonstrated breakthroughs in realism and controllability for multi-agent systems. We will revise our Related Work section to include a summary of this discussion.
>
> [1] Denoising diffusion probabilistic models. (NeurIPS 2020) \
> [2] Argmax flows and multinomial diffusion: Learning categorical distributions. (NeurIPS 2021) \
> [3] Structured denoising diffusion models in discrete state-spaces. (NeurIPS 2021) \
> [4] Generalized interpolating discrete diffusion. (ICML 2025)

---

> ### Author Response · Authors · 2025-11-19
> **Part (2/2)**
>
> ### Architectural Design (CrossGuid)
> The reviewer is correct that CrossGuid utilizes cross-attention. However, its innovation lies in its purpose-built design for multi-agent conditioning, not in the mechanism itself.
>
> Unlike image diffusion where guidance can be added to aligned feature maps, our problem involves fusing a guidance sequence (length L) with a spatio-temporal agent tensor (T×N). Cross-attention is the minimal, most principled operation for this structural alignment.
>
> Rather than being "standard engineering," CrossGuid represents the simplest possible solution to enable our novel control tasks within the joint diffusion framework. Its implementation is specifically tailored for multi-agent domains, with distinct designs for possessor sequences (updating only the ball) and text guidance (updating all agents), making it an essential enabler rather than arbitrary complexity.
>
>
> ### Consistent Generation
> We thank the reviewer for this insightful question. This is a crucial point, and we can confirm that our joint model produces highly consistent outputs. We verified this in two ways:
> * Qualitative: As shown in our supplementary video, the predicted possessor (highlighted in yellow) visually aligns with the player who has the ball, demonstrating a tight, frame-level coupling. Furthermore, the end of the supplementary video shows the ball's attention mechanism correctly concentrating on the players who will take possession, further supporting that the model learns the joint-distribution.
> * New Quantitative Analysis: We provide a new experiment to explicitly measure consistency of the joint generation $\hat{X}_0 = (\hat{Y}_0, \hat{E}_0)$. Using the same setup as in Table 2, we generate 20 samples per scene and compute the Acc metric between the predicted discrete possessor $\hat{E}_0$ and the threshold-based heuristic possessor extracted from the predicted trajectories $\hat{Y}_0$. For reference, we compare our **multinomial** diffusion with the commonly used **absorbing**-state formulation (Levi et al. 2023, Li et al. 2025). Unlike the multinomial model, which enables refinement through all states at each denoising step, the absorbing-state mechanism freezes predicted variables once unmasked, preventing later refinement [4]. As shown below our method achieves high consistency, particularly for the best sample ($max$), and strong consistency on average ($avg$):
>
> | Method                   | NFL (Multinomial) | Bundesliga (Multinomial) | NBA (Multinomial) | NFL (Absorbing) | Bundesliga (Absorbing) | NBA (Absorbing) |
> |--------------------------|-------------------|---------------------------|--------------------|------------------|--------------------------|------------------|
> | w/o $\mathcal{G}$        | **.98 / .86**     | **.97 / .80**             | **.99 / .92**      | .97 / .80        | .94 / .70                | **.99** / .89   |
> | w $\mathcal{G}_\text{WPG}$     | **.96 / .84**     | **.95 / .80**             | **.99 / .92**      | .93 / .78        | .91 / .68                | **.99** / .90    |
> | w $\mathcal{G}_\text{text}$    | **.97 / .86**     | **.96 / .81**             | -                  | .95 / .81        | .95 / .76                | -                |
>
> The variance in performance for NFL and Bundesliga correlates with the smaller training set sizes. However, it is crucial to note that these discrepancies are mostly minor temporal offsets instead of incorrect predictions. The model consistently identifies the correct player but occasionally lags in precise frame alignment (e.g., see min 2:11 in the video, where Player 19 is correctly identified with a slight temporal delay). Furthermore,  we observe an average Acc drop of ~6% on NFL, ~10% on Bundesliga and ~2-3% on NBA compared to our multinomial proposal. This ablation demonstrates that multinomial diffusion provides more consistent event–trajectory predictions than absorbing-state diffusion.
>
> We will add this full consistency analysis, including the new table, to the final version of the paper to address this consistency point.
>
> We hope these clarifications and the new analysis fully address the reviewer's concerns regarding novelty and soundness.

---

### Official Review · Reviewer_nJy6 · 2025-11-05

**Soundness:** 3
**Presentation:** 3
**Contribution:** 2
**Rating:** 4
**Confidence:** 4

**Summary:**

This paper introduces JointDiff, a diffusion framework for multi-agent trajectory generation in sports. The framework is designed to unify continuous and discrete diffusion processes to simultaneously generate continuous spatiotemporal trajectories and discrete events (e.g., possession of ball). The paper presents two controllable generation scenarios (weak-possessor-guidance and text-guidance). The paper reports strong performance on scene-level metrics.

**Strengths:**

The paper's core hypothesis—that joint modeling of continuous trajectories and discrete events is beneficial—is validated on the domains of interest, as evaluated by scene-level metrics (Table 1).  The paper also successfully demonstrates two forms of semantic control (WPG and text-guidance -- shown in Table 2).  The paper also contributes a new text-annotated benchmark for the NFL and BundesLiga datasets, which enables the text-guidance task.

**Weaknesses:**

The authors claim this is the "first to introduce this joint modeling paradigm to the domain of time-evolving multi-agent systems". This claim, while true in its most narrow interpretation, overlooks a substantial and directly relevant body of un-cited prior art in adjacent fields like robotics and autonomous driving, where joint discrete-continuous diffusion models are already an active area of research for dynamic planning and generation.  (The authors do cite MotionDiffuser.)

I would strongly encourage the authors to engage with the broader literature in these adjacent fields:
https://arxiv.org/abs/2509.21983
https://sites.google.com/view/dimsam-tamp
https://generative-skill-chaining.github.io/
https://arxiv.org/abs/2509.20109
https://arxiv.org/abs/2405.15677
https://arxiv.org/abs/2402.11502
My goal in listing these papers is not to say that any of them are doing exactly what the authors are doing.  But in characterizing the specific novelty of this paper, it is important to situate it in this broader context.

The paper's architecture appears potentially over-engineered, consisting of a complex stack of modules (Temporal Mamba, Social Transformers, and CrossGuid) built upon the U2Diff architecture. The authors list limitations such as the "architecture requires events to share the same spatio-temporal structure as the trajectory data", but to me this type of limitation is a by-product of the engineering design.

**Questions:**

I would like the authors to consider the broader landscape of related work, and then clarify what they view as the core novelty of this paper.

---

> ### Author Response · Authors · 2025-11-20
> **Part (1/2)**
>
> We thank the reviewer for their valuable feedback and for providing a list of related works, which has helped us clarify the precise contribution of our paper. We address their concerns below.
>
> ### Core Novelty
> Our principal contribution is JointDiff, the first framework to successfully unify discrete and continuous diffusion for the generative modeling of dynamic, multi-agent systems. This joint formulation not only establishes a new state-of-the-art for standard generative tasks but also unlocks novel semantic control capabilities, such as text-guided generation. To support this, we introduce a unified benchmark with text annotations.
>
> The core of our approach is a unified generative framework that synchronously applies a continuous diffusion process (DDPM [1]) to trajectories and a discrete, multinomial diffusion process [2] to events. This enables mutual, bidirectional conditioning: the denoising of trajectories is informed by the evolving event state, and in reverse. We demonstrate that this co-evolution is a key differentiator, leading to more realistic generations (Table 1), enabling new forms of control (Table 2), and resulting in outputs preferred by human evaluators.
>
> [1] Denoising diffusion probabilistic models. (NeurIPS 2020) \
> [2] Argmax flows and multinomial diffusion: Learning categorical distributions. (NeurIPS 2021)
>
>
> ### Relation to Prior Work in Robotics and Autonomous Driving
> We thank the reviewer for pointing to this broader landscape [3-8] (please refer to the **Note on non-contemporaneous work** in Part 2/2). We agree that situating JointDiff in this context is crucial and will add a discussion of these works to our Related Work section. We have analyzed the cited papers and find they differ from JointDiff in fundamental ways:
> * **Multi-agent vs. Single-agent:**
> The robotics works [3-5] and the autonomous driving (AD) work [6, 8] focus on single-ego planning (a single robot's motion or a single ego-car's actions). JointDiff is explicitly designed for multi-agent systems, where the interactions, coordination, and global consistency between all agents (N=23 in football / soccer, N=11 in basketball) are the primary challenge.
> * **Non-diffusion baselines:**
>  GenAD [6] and SMART [7] are non-diffusion based baselines, representing a different generative paradigm altogether.
> * **(Continuous AND Discrete) vs. (Continuous OR Discrete):**
> Works [3, 4] model variables as continuous (robot state, action parameters, trajectories) using standard DDPM, without a parallel discrete diffusion process.
> The AD work [8] discretizes trajectories into codebooks, modeling them as discrete variables, to apply discrete denoising diffusion without continuous. However, JointDiff is designed to handle the intrinsic mixed nature of the data directly: it uses a continuous process for trajectories to preserve spatial precision and a discrete process for events to capture categorical changes.
> * **Nature of the discrete process:**
> The non-contemporaneous work [5] is the only using joint continuous-discrete (hybrid) diffusion coupling DDPM with discrete diffusion. However, as in [8], they use an absorbing-state discrete diffusion (mask → token transitions) with cross-entropy loss. This framework sacrifices variable-level refinement after it has been unmasked [16]. Please refer to [15] for further information of the type of discrete diffusion models. JointDiff uses multinomial diffusion (defined as uniform in [15]), where pure noise is a uniform categorical distribution, and training uses the full $\mathcal{L}_{vb}$ objective. It provides fine-grained token-level refinement, allowing the model to make changes across the entire sequence at any step, which is vital for simultaneous generation of complex, long-horizon sequences. For quantitative comparison, refer to the **Consistent Generation** analysis in Reviewer 5jzA, where we compare our multinomial against the absorbing-state framework.
>
> To summarize: JointDiff demonstrates joint discrete-continuous diffusion for the general task of multi-agent trajectory completion while enabling high-level semantic controllability. We will update the Related Work section to add the contemporaneous references [3, 4, 6, 7].
>
> [3] Generative Skill Chaining: Long-Horizon Skill Planning with Diffusion Models (CoRL 2023) \
> [4] DiMSam: Diffusion Models as Samplers for Task and Motion Planning under Partial Observability (IROS 2024) \
> [5] Hybrid Diffusion for Simultaneous Symbolic and Continuous Planning (arxiv, submitted on **26 Sep 2025**) \
> [6] GenAD: Generative End-to-End Autonomous Driving (ECCV 2024) \
> [7] SMART: Scalable Multi-agent Real-time Motion Generation via Next-token Prediction (NeurIPS 2024) \
> [8] Discrete Diffusion for Reflective Vision-Language-Action Models in Autonomous Driving (arxiv, submittted on **24 Sep 2025**) \
> [15] Structured denoising diffusion models in discrete state-spaces (NeurIPS 2021) \
> [16] Generalized interpolating discrete diffusion (ICML 2025)

---

> > ### Author Response · Authors · 2025-11-20
> > **Part (2/2)**
> >
> > (**Note on non-contemporaneous work**: We note that the provided [5] and [8] were made public on arXiv on 26th and 24th September, respectively. These dates correspond to the submission deadline of this venue. While we are happy to discuss them, we believe they fall outside the 30-day window for required comparison per ICLR guidelines and should not be used to assess our submission. Nonetheless, we have included detailed technical comparisons above to clarify our distinct contributions.)
> >
> > ### Architectural Design
> > We appreciate the reviewer's perspective on the architecture's complexity. We argue that our design is a principled decomposition of the multi-agent generation problem, where each module addresses a distinct, well-defined challenge:
> > * Temporal Mamba [14]: Efficiently models long-range temporal dependencies for each agent (an O(T) complexity problem for each agent).
> > * Social Transformers [14]: Captures complex inter-agent interactions at each timestep (an O(N²) complexity problem for each timestep).
> > * CrossGuid: Provides a minimal interface for injecting high-level semantic guidance (text or WPG) into the multi-agent state.
> >
> > The decoupling of temporal and social dynamics within the Social–Temporal Block follows a well-established practice in multi-agent modeling [9, 10, 13, 14], shown to improve efficiency and flexibility. Overall, our architecture can be viewed as a streamlined adaptation of general diffusion-based frameworks such as [12, 13, 14], customized for joint discrete–continuous generation. CrossGuid, based on cross-attention, is the simplest and most principled operation for fusing guidance signals (of length L) with our multi-agent tensor (of shape T×N), which have inherently different sequential structures.
> >
> > [9] Latent variable sequential set transformers for joint multi-agent motion prediction. (ICLR 2022) \
> > [10] Scene transformer: A unified architecture for predicting multiple agent trajectories. (ICLR 2022) \
> > [11] Transportmer: A holistic approach to trajectory understanding in multi-agent sports. (ACCV 2024) \
> > [12] Diffwave: A versatile diffusion model for audio synthesis. (ICLR 2021) \
> > [13] Csdi: Conditional score-based diffusion models for probabilistic time series imputation. (NeurIPS 2021) \
> > [14] Unified uncertainty-aware diffusion for multi-agent trajectory modeling. (CVPR 2025)
> >
> > **Reframing the limitation: a deliberate scope for a fundational contribution**
> > We acknowledge the architectural constraint requiring synchronous event streams. This was a deliberate design choice to create a tractable framework for validating our core contribution: that joint modeling itself significantly improves multi-agent generation. By first solving the problem for dense events, we establish a strong foundation and clear baseline. Extending this to sparse, asynchronous events via temporal point processes represents a natural and valuable direction for future work built upon this proven foundation.
> >
> > We hope these clarifications and our detailed comparison to the broader literature properly situate the novelty of JointDiff.

---

### Author Response · Authors · 2025-12-02
**Summary of the Discussion**

Dear Area Chairs,

The reviewers recognize this paper as addressing an "important yet challenging problem" in controllable multi-agent generation (nw5t) and presenting the "first work to introduce joint continuous & discrete diffusion modeling" for this task (5jzA), successfully validating its "core hypothesis" (nJy6). The method demonstrates "two forms of semantic control" (nJy6) with "new guidance methods" (5jzA) and contributes a "new text-annotated benchmark" (nJy6). Supported by "comprehensive" evaluation and a "human study" (5jzA), its "principled" formulation (Q4Be) and "very clear" architecture (Q4Be) provide a framework that "enables interesting applications" (Q4Be) and "can be extended to many other large-scale control systems" (nw5t).


To address the concerns raised in the reviews, we summarize the key revisions and clarifications:
- **Clarified Novelty & Related Work**: Following Reviewer nJy6, we have incorporated the provided **contemporaneous** references [3,4,6,7] into the **Related Work** section, noting that none employ joint continuous-discrete diffusion. We further discuss the differences between our aproach and these in our rebuttal. In response to Reviewer 5jzA, we have revisited the **Related Work** section to provide a rationale comparison that distinguishes our contribution from prior joint-diffusion methods. This comparison is supported by an extended qualitative analysis and comparison table in the rebuttal discussion. We have also strengthened our second contribution to emphasize our work's role in "enabling high-level semantic controllability in dynamic domains", with a particular focus on our complex multi-agent setting.

- **CrossGuid design**: Per Reviewers nJy6 and 5jzA, we clarified that CrossGuid is intentionally designed as the simplest yet effective cross-attention–based mechanism for controllability in multi-agent domains, mapping a source signal of length $L$ to a structured multi-agent embedding of size $T \times N$. This design is essential for demonstrating controllability within joint diffusion.

- **Enhanced Analysis & Ablations**: Per Reviewer 5jzA, a new consistency analysis and ablation study comparing our formulation to the absorbing state baseline has been added (**Section 5.3, Appendix D.3.5**). Per Reviewer Q4Be, we expanded comparisons (**Appendix D.2.4**), added further ablations (**Appendices D.3.3, D.3.4**), and provided additional clarifications based on their thoughtful questions.

- **Improved Accessibility**: Following Reviewer Q4Be, the human evaluation is now included in the main paper (**Section 5.1**). Per Reviewer nw5t, we revised all figures for clarity with increased font sizes, improved the visual correspondence between **Figure 2** and its extended version **Figure 5** (Figure 4 before rebuttal), and refined trajectory visualizations to better highlight key agents.


After the rebuttal, we received one updated reviewer feedback before 21 November:
- Reviewer Q4Be increased the score from 6 to 8.


Best, \
Authors of submission #9796

---

### Meta-Review · Area_Chair_CKwf · 2026-01-05

**Summary:**

The main concerns of the reviewer are related to the novelty and how the paper is positioned in relation to related works. With multiple reviewers pointing to highly relevant fields working on joint discrete-continuous diffusion in various data domains [nJy6, 5jzA, Q4Be]. The paper would benefit from having a more comprehensive discussion of related work [nJy6, Q4Be]. Also, reviewers pointed out how the design seems a bit complex with multiple components [nJy6, 5jzA, nw5t]. There are also miscellaneous regarding font size and how some illustrations are not as clear [nw5t]

**Reviewer Concerns:**

After reading the rebuttal and reviewers’ comments. The AC believes that most of the concerns were addressed. Specifically, the details are provided, clarifications were made to the method, and additional comparisons are provided. The more difficult concern to decide is whether the related works are adequately addressed. Beyond the already mentioned references, the AC also thinks that the work could benefit from discussing its relevance to methods on sports analytics, e.g.,
- Zheng, Stephan, Yisong Yue, and Jennifer Hobbs. "Generating long-term trajectories using deep hierarchical networks." Advances in Neural Information Processing Systems 29 (2016).
- Yeh, Raymond A., et al. "Diverse generation for multi-agent sports games." Proceedings of the IEEE/CVF Conference on Computer Vision and Pattern Recognition. 2019.
- Alcorn, Michael A., and Anh Nguyen. "baller2vec: A multi-entity transformer for multi-agent spatiotemporal modeling." arXiv preprint arXiv:2102.03291 (2021).
- Mao, Weibo, et al. "Leapfrog diffusion model for stochastic trajectory prediction." Proceedings of the IEEE/CVF conference on computer vision and pattern recognition. 2023.

**Reviewer Scores:**

In the AC opinion, if the reviewers were able to fully participate in the discussion, the concerns of  [nJy6, 5jzA] regarding related works could be resolved. Additionally, Reviewer Q4Be commented that they would increase the score to 8, and Reviewer nw5t only had minor concerns.

Misc:
The paper title for "Yi Xu and Yun Fu. Deciphering movement: Unified trajectory generation model for multi-agent. arXiv preprint arXiv:2405.17680, 2024" has been updated on arXiv.

---

### Decision · Program_Chairs · 2026-01-26

Accept (Poster)